# THE PATH OF LEAST RESISTANCE: GUIDING LLM REASONING TRAJECTORIES WITH PREFIX CONSENSUS

**Ishan Jindal**[*]
Fujitsu Research India
ishan.jindal@fujitsu.com

**Sai Prashanth Akuthota†, Jayant Taneja†, Sachin Dev Sharma**
Samsung R&D Institute India-Delhi
{a.prashanth, j.taneja, sachin.dev}@samsung.com

## ABSTRACT

Large language models achieve strong reasoning performance, but inference strategies such as Self-Consistency (SC) are computationally expensive, as they fully expand all reasoning traces. We introduce **PoLR** (*Path of Least Resistance*), the first inference-time method to leverage *prefix consistency* for compute-efficient reasoning. PoLR clusters short prefixes of reasoning traces, identifies the dominant cluster, and expands all paths in that cluster, preserving the accuracy benefits of SC while substantially reducing token usage and latency. Our theoretical analysis, framed via mutual information and entropy, explains why early reasoning steps encode strong signals predictive of final correctness. Empirically, PoLR consistently matches or exceeds SC across GSM8K, MATH500, AIME24/25, and GPQA-DIAMOND, reducing token usage by up to 60% and wall-clock latency by up to 50%. Moreover, PoLR is fully complementary to adaptive inference methods (e.g., Adaptive Consistency, Early-Stopping SC) and can serve as a drop-in pre-filter, making SC substantially more efficient and scalable without requiring model fine-tuning.

## 1 INTRODUCTION

Large Language Models (LLMs) have recently achieved remarkable performance on complex reasoning tasks Grattafiori et al. (2024); Yang et al. (2025); Guo et al. (2025); Jindal et al. (2025), ranging from grade-school math (Cobbe et al., 2021) to graduate-level problem solving (Hendrycks et al., 2021; Rein et al., 2023). Among inference-time strategies, *Self-Consistency* (SC) decoding (Wang et al., 2023) has emerged as a strong default: by sampling multiple reasoning traces and taking a majority vote over their final answers, SC substantially improves accuracy over greedy or single-sample decoding. However, it incurs substantial computational cost because all reasoning traces must be expanded to completion.

To reduce SC's compute requirements, several inference-time methods such as Adaptive Consistency (AC) Aggarwal et al. (2023) and Early-Stop Self-Consistency (ESC) Li et al. (2024) have been proposed. These methods expand reasoning traces sequentially and stop generating them only when sufficient final-answer agreement is observed. Though effective, they share a fundamental limitation: answer-level agreement is only observable *after* full reasoning traces is generated. As a result, they cannot exploit the rich structural information that might appear earlier in the reasoning process and their efficiency remains limited by the need to generate complete reasoning traces.

Recently, an alternative line of research shows that the early stages of reasoning traces carry disproportionately strong signals about the eventual solution, a phenomenon known as *prefix consistency*. Formally, if $r_i$ denotes a reasoning trace, then its first $L$ tokens $r_i[1:L]$, termed as prefix, tend to exhibit similarity across reasoning traces, irrespective of their later steps. Ji et al. (2025) exploited this phenomenon at *training time*, that is, fine-tuning models on prefixes to improve reasoning while

---

[*]Work done while at SRID, †Contributed equally

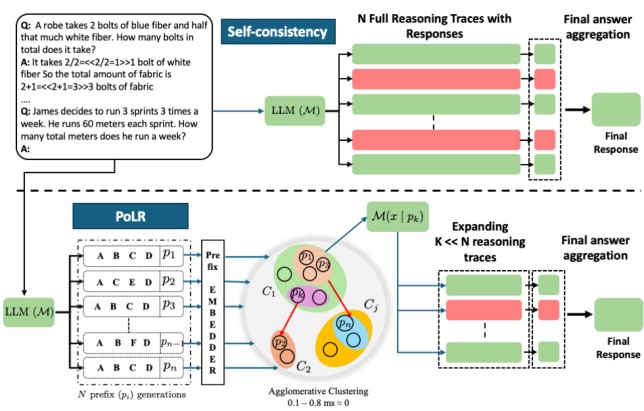

(a) PoLR Overview

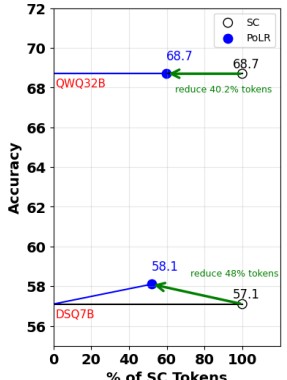

(b) Token-accuracy plot GPQA-Diamond

Figure 1: (a) Comparison of Self-Consistency (SC) and PoLR. **Top:** SC expands all $N$ sampled traces to completion (100% expansion), then aggregates answers via majority vote. **Bottom:** PoLR first generates $N$ short prefixes of length $L_p$, embeds and clusters them, and selects the dominant cluster. All $K \ll N$ traces from this cluster are expanded, after which majority voting is applied. (b) PoLR exceeds SC accuracy, while reducing token cost by approx 50%.

reducing inference cost. However, this requires expensive fine-tuning and cannot be applied directly at inference.

This gap motivates a method that reduces Self-Consistency cost by exploiting early steps of reasoning traces rather than waiting for full trajectories. To address this need, we introduce **PoLR** (*Path of Least Resistance*), the first method to leverage prefix consistency for *inference-time Self-Consistency*. To leverage Prefix consistency, PoLR generates $N$ short prefixes, embeds and clusters them, and only expands the prefixes to full reasoning traces for the dominant cluster, reducing wasted token generation and adaptively allocating compute to promising paths. This approach preserves SC's accuracy while cutting token usage and latency dramatically as depicted in Figure 1. The key contributions of this work are:

- PoLR is a drop-in SC replacement that clusters partial reasoning traces and selectively expands the dominant cluster, substantially reducing inference cost.
- Across math (GSM8K, MATH500, AIME24/25), commonsense, and science reasoning benchmarks (GPQA-DIAMOND), and implicit knowledge retrieval benchmark (STARTEGYQA) PoLR shows up to 60% token reduction and 50% latency savings without accuracy loss, consistent across LLM families and scales (1–32B params).
- PoLR is the first inference-time method to exploit prefix consistency for Self-Consistency, complementing existing adaptive self-consistency methods further reducing the token generation.
- PoLR is robust to different clustering methods, prefix lengths, and cluster selection strategies.

## 2 DO PREFIXES ENCODE EARLY CONSENSUS?

To understand if prefixes encode strong signals about the eventual solution, we conduct a preliminary analysis of reasoning prefixes. We generate 40 traces per question using **DeepSeek-Distill-Qwen-7B** (DSQ7B) on MATH500 and GSM8K, truncating traces at varying prefix lengths $L_p$ in Table 1. We evaluate the fraction of traces that shares identical prefix $L_p$ (**Expansion rate**) and compute majority-vote (**accuracy**). We also find the **exact prefix match (EPM)** that is number of problems where all 40 traces share identical prefixes. The results show that traces sharing the same prefix achieve nearly the same ac-

Table 1: Preliminary analysis on MATH500 and GSM8K (DSQ7B, 40 samples).

|  | $L_p$ | Expansion rate | Accuracy | EPM |
|---|---|---|---|---|
|  | SC | 1.00 | 89.8 | – |
| MATH500 | 32 | 0.64 | 89.8 | 125 |
|  | 64 | 0.58 | 89.6 | 63 |
|  | 128 | 0.48 | 89.2 | 5 |
|  | 256 | 0.45 | 89.2 | 0 |
|  | SC | 1.00 | 79.7 | – |
| GSM8K | 32 | 0.52 | 79.7 | 135 |
|  | 64 | 0.49 | 78.9 | 80 |
|  | 128 | 0.47 | 79.3 | 30 |
|  | 256 | 0.46 | 79.1 | 1 |

curacy as full SC, meaning a large fraction of token cost spent on extra traces rarely contribute to the final answer. These findings suggest that LLMs encode structural agreement well before generating complete answers. Detecting and leveraging this early consensus can substantially reduce compute without compromising SC's robustness.

## 3 PATH OF LEAST RESISTANCE (PoLR)

In this section, we present how *PoLR (Path of Least Resistance)* operates as an inference-time alternative to Self-Consistency (SC). The name PoLR is motivated by a natural principle: systems tend to follow the *path of least resistance*, avoiding unnecessary detours while conserving energy. Analogously, instead of fully expanding all reasoning traces like SC, PoLR prunes unlikely or redundant paths early and allocates computation only to the dominant prefix cluster.

### 3.1 MATHEMATICAL FORMULATION

**Setup.** Consider an input reasoning question $x$ posed to a language model $\mathcal{M}$. In the standard SC baseline, we sample $N$ complete reasoning traces of average length $\ell_f$ and then aggregate their final answers by majority vote. While effective, this incurs high inference cost because every trace must be expanded until completion, even if most are redundant.

PoLR modifies this pipeline by introducing a prefix-based selection step. Specifically, we sample $N$ short prefixes, each of length $L_p$ tokens, and use their semantic structure to decide which subset of traces to fully expand. The key intuition is that reasoning traces often share overlapping initial steps (Ji et al., 2025), and these early structures correlate with the correctness of the final outcome. By clustering prefixes, we can identify the dominant reasoning mode early, without generating all traces to completion. A step-by-step implementation instructions presented in Appendix B (Algorithm 1).

**Step 1: Prefix Sampling.** Given the input question $x$ and the $\mathcal{M}$, we first generate $N$ short reasoning prefixes $p_i = \text{Prefix}(\mathcal{M}(x, t_i), L_p)$, $\quad i = 1, \ldots, N$, where $t_i$ is the sampling temperature, and $\text{Prefix}(\cdot)$ denotes truncating the LLM output to $L_p$ tokens. In practice, this is implemented by setting `max_new_tokens = ` $L_p$.

**Step 2: Embedding and Clustering.** Each prefix $p_i$ is embedded into a sparse vector representation via *TF–IDF bag-of-words encoding* over tokens. This choice is lightweight, model-agnostic, and CPU-friendly, avoiding external neural encoders. We provide a detailed comparison with neural encoder in Table 4. It is evident that neural encoders increase the clustering overhead way more than the TF-IDF encoders with diminishing returns on the accuracies.

We cluster $\{p_i\}$ into $\mathcal{C} = \{C_1, \ldots, C_m\}$ using *Agglomerative Hierarchical Clustering* with cosine similarity. This is well-suited for small $N$ (11–51), as it requires no pre-specified $m$ and produces interpretable groupings. That is $C^* = \arg\max_{C_j \in \mathcal{C}} |C_j|$, where $\bigcup_{j=1}^{m} C_j = \{p_1, \ldots, p_N\}$ and $C^*$ is the dominant cluster.

**Step 3: Expansion.** We then expands all $K$ prefixes from $C^*$ to full reasoning traces as $r_k = \mathcal{M}(x \mid p_k)$, $\quad p_k \in C^*$, $\quad k = 1, \ldots, K$.

**Step 4: Self-Consistency Voting.** Let $a_k$ be the extracted answer from trace $r_k$. PoLR returns $\hat{a} = \arg\max_y \sum_{k=1}^{K} \mathbf{1}[a_k = y]$. Thus PoLR strictly generalizes SC: if $K = N$ and clustering is bypassed, it reduces to standard SC.

**Token Efficiency:** Let $\ell_p$ = average prefix length, $\ell_f$ = full reasoning length. Number of tokens generated for SC $T_{\text{SC}} = N \cdot \ell_f$, and for PoLR $T_{\text{PoLR}} = N \cdot \ell_p + K \cdot (\ell_f - \ell_p)$. We compute the token efficiency as:

$$\eta = 1 - \frac{T_{\text{PoLR}}}{T_{\text{SC}}} = 1 - \frac{N \cdot \ell_p + K \cdot (\ell_f - \ell_p)}{N \cdot \ell_f}.$$

## 3.2 THEORETICAL JUSTIFICATION

PoLR relies on the intuition that early prefixes already contain useful signals about the final reasoning trajectory. We formalize this intuition by considering two complementary properties: (i) correctness alignment, which determines whether restricting to a dominant cluster preserves accuracy, and (ii) structural skew, which governs the magnitude of efficiency gains.

### 3.2.1 CORRECTNESS ALIGNMENT AND ACCURACY PRESERVATION

Let $Y \in \{0, 1\}$ denote the correctness of a final reasoning trajectory (1 if correct, 0 otherwise), and let $Z$ denote the cluster assignment of a sampled prefix. The critical condition for PoLR is that $Z$ carries information about $Y$, i.e. $I(Z; Y) > 0$, where $I(\cdot; \cdot)$ denotes mutual information. Intuitively, if prefixes cluster in a way that is at least weakly predictive of correctness, then restricting expansion to the dominant cluster will not systematically degrade accuracy. In this view, self-consistency (SC) can be seen as an unbiased estimator of $\mathbb{E}[Y]$, while PoLR acts as a variance-reduced estimator that focuses on high-probability clusters.

Formally, the conditional entropy of correctness given the cluster assignment can be written as $H(Y|Z) = \sum_z P(Z = z)H(Y|Z = z)$. If $H(Y|Z)$ is small, then cluster identity reliably predicts correctness. Our empirical results (Section 5) show that $I(Z; Y)$ and $H(Y|Z)$ remain non-trivial across models, which explains why PoLR consistently matches SC in accuracy.

### 3.2.2 STRUCTURAL SKEW AND EFFICIENCY

While correctness alignment governs accuracy preservation, our experiments reveal that it does not explain the magnitude of efficiency gains. Instead, efficiency is driven by the *structural skew* in the prefix cluster distribution. Define the skew for a given instance as $\kappa = \frac{|C^*|}{N}$, where $|C^*|$ is the size of the dominant cluster and $N$ is the number of sampled prefixes. If $\kappa$ is large, the majority of prefixes fall into one cluster, and ignoring the smaller clusters eliminates substantial redundant expansions. Conversely, if clusters are balanced ($\kappa \approx 1/m$), dominant cluster's traces' expansion yields more token savings but poorer quality.

At the dataset level, the expected efficiency gain is thus directly tied to the expected skew $\mathbb{E}[\eta] \propto \mathbb{E}[\kappa^{-1}]$. Empirically, we observe strong correlation between $\kappa$ and token savings, whereas NMI between clusters and correctness is weakly correlated with efficiency. This indicates that PoLR's efficiency derives from structural dominance rather than correctness alignment.

The combined picture is as follows:

- **Accuracy preservation** requires that $I(Z; Y) > 0$, i.e., that clusters are not adversarially misaligned with correctness. Even modest alignment is sufficient, as SC's voting ensures that errors do not amplify.
- **Efficiency magnitude** depends on structural skew $\kappa$: the more dominant the largest cluster, the more redundant expansions PoLR can safely ignore.

This separation of concerns reconciles our theory and empirical findings: mutual information guarantees safety, while skew determines savings. Our experiments across GSM8K with multiple models (1.5B–7B) confirm this in Section 5.5, where NMI remains low ($\leq 0.18$), yet efficiency saturates around 50–58%, precisely because prefix clusters exhibit strong structural skew.

We now make the connection between cluster skew and efficiency gains explicit.

**Proposition 1.** *Let $N$ denote the number of sampled prefixes, partitioned into $m$ clusters $\{C_1, \ldots, C_m\}$ with sizes $|C_1|, \ldots, |C_m|$, and let $C^*$ denote the dominant cluster with size $|C^*|$. Assume PoLR expands $K$ continuations from $C^*$, while Self-Consistency (SC) expands $M$ continuations from all $N$ prefixes (with $M \geq K$). Then the expected token efficiency gain of PoLR relative to SC satisfies*

$$\eta \;\geq\; 1 - \frac{K}{M} \cdot \kappa^{-1}, \text{where } \kappa = \frac{|C^*|}{N} \text{ is the dominance ratio (skew).}$$

*Sketch.* SC requires expanding $M$ continuations distributed across all $N$ prefixes. If $m$ clusters are expanded proportionally, each prefix contributes on average $m/N$ expansions. PoLR instead

expands only $K$ continuations from the dominant cluster $C^*$. Normalizing by $M$, the relative cost is $\frac{K}{M} \cdot \frac{N}{|C^*|} = \frac{K}{M} \cdot \kappa^{-1}$. Thus the efficiency gain relative to SC is at least $1 - \frac{K}{M} \cdot \kappa^{-1}$. Equality holds when expansions are exactly proportional across clusters. $\qquad\square$

This bound formalizes the empirical observation that efficiency gains scale monotonically with $\kappa$: the more dominant the largest cluster, the more redundant expansions can be ignored.

## 4 MAIN EXPERIMENTS

**Backbone LLMs.** We evaluate the efficiency and generality of *PoLR – Path of Least Resistance* across diverse open-source LLMs spanning different architectures, scales, and training paradigms. Specifically, we use **DeepSeek-R1-Distill-Qwen (DSQ)** (7B, 1.5B) (Guo et al., 2025), distilled from reasoning-specialized LLMs; **QWQ32B** (Team, 2025; Yang et al., 2024a), a Qwen2.5 variant trained with reinforcement learning for problem solving; **MiMo-7B-RL-0530** (Xiaomi, 2025) and **Phi-4-15B** Abdin et al. (2025), an open-source GPT-style model trained with large-scale supervised data; and **Qwen2.5-Math-7B** (Yang et al., 2024b), a math-specialized instruction-tuned model. These choices cover architectures (Qwen, MiMo, Phi-4, DeepSeek), parameter scales (1.5B–32B), and training paradigms (distillation, RL, supervised fine-tuning).

**Benchmarks.** We evaluate on multi-step reasoning tasks: GSM8K (Cobbe et al., 2021), grade-school arithmetic word problems; MATH500 (Lightman et al., 2023), a set of 500 challenging math problems; AIME24/25 (of Problem Solving, 2024; 2025), high-school olympiad-level math problems; GPQA-DIAMOND (Rein et al., 2024), a graduate-level STEM reasoning benchmark covering physics, chemistry, and biology; and STRATEGYQA (Geva et al., 2021), a multi-hop reasoning and implicit knowledge retrieval task.

**Evaluation Metrics.** We follow standard metrics from prior reasoning literature: **Exact Match (EM)** on GSM8K (Habib et al., 2023); **Pass@1** on Math500 and AIME24/25; **Accuracy** (binary correctness) on GPQA-Diamond. We also measure **Token Efficiency** relative to Self-Consistency (SC): $\eta = 1 - \frac{T_{\text{PoLR}}}{T_{\text{SC}}}$, **Path Expansion (PExp)** denoting the number of full reasoning traces used for majority voting, and **PoLR Overhead** ($k_t$), which includes TF-IDF vectorization and clustering.

**Baselines.** The primary baseline is **Self-Consistency (SC)** (Wang et al., 2023), which samples multiple chain-of-thoughts independently and selects the majority answer. SC is widely adopted as a standard inference-time ensemble for reasoning tasks. We also report single-sample greedy decoding (Chain-of-Thought, CoT) as a lower-bound reference and compare PoLR with **Adaptive Consistency (AC)** Aggarwal et al. (2023), and **Early-Stopping Self-Consistency (ESC)** Li et al. (2024).

All experiments are repeated 10 times[1]. with different random seeds (sampling order and temperature) and we report mean performance and standard deviation across runs for all metrics: accuracy, EM, Pass@1, token efficiency, and latency. Further hyperparameter details are provided in Appendix A. All PoLR evaluations use $L_p = 256$ unless stated otherwise. Empirically, we find that $L_p = 256$ achieves a good balance between PoLR accuracy and token efficiency gains.

**Main Results.** Table 2 presents the performance of **PoLR** compared to Self-Consistency (SC) across five reasoning benchmarks (GSM8K, MATH500, AIME24, AIME25, GPQA-DIAMOND) and multiple LLM families. The results reveal clear advantages of PoLR. First, PoLR drastically reduces token usage while preserving accuracy. Across all datasets and models, token efficiency $\eta$ typically ranges between **40–60%**, effectively cutting token consumption by roughly half. For example, on GSM8K with QWQ32B at $N = 51$, PoLR achieves the same accuracy as SC (**90.8%**) while using only half the tokens ($\eta = 47.6\%$). The additional clustering overhead $k_t$ is minimal, just a few milliseconds, so the savings directly translate into faster inference. Second, accuracy is preserved and occasionally improved. Despite discarding up to half of the reasoning paths, PoLR matches SC's accuracy and sometimes surpasses it. On MATH500, for instance, PoLR improves

---

[1]For brevity, standard deviations are not included in the main table. All reported gains are statistically significant. Detailed standard deviation values are provided in Appendix C.

Table 2: Performance comparison of PoLR versus SC across datasets (GSM8K, MATH500, AIME24, AIME25, GPQA-DIAMOND) and model sizes. The table shows accuracy differences (green = improvement, red = drop), token efficiency $\eta$ (%), sample size $N$, and PoLR overhead $k_t$ (ms).

| | | DSQ1.5B | | | | DSQ7B | | | | QWQ32B | | | |
|---|---|---|---|---|---|---|---|---|---|---|---|---|---|
| GSM8K | $N$ | SC | PoLR | $\eta(\%)$ | $k_t$ (ms) | SC | PoLR | $\eta(\%)$ | $k_t$ (ms) | SC | PoLR | $\eta(\%)$ | $k_t$ (ms) |
| (1319) | 51 | 73.2 | 0.0 | 40.1 | 11.3 | 79.8 | 0.2 | 26.5 | 5.9 | 90.8 | -0.3 | 47.6 | 11.2 |
| | 31 | 73.2 | 0.0 | 41.1 | 6.0 | 80.0 | -0.4 | 27.2 | 4.3 | 90.9 | -0.6 | 48.6 | 5.8 |
| | 11 | 72.6 | 0.0 | 43.7 | 2.3 | 79.7 | 0.0 | 28.1 | 1.9 | 90.6 | -1.3 | 54.2 | 2.4 |

| | | DSQ1.5B | | | | DSQ7B | | | | QWQ32B | | | |
|---|---|---|---|---|---|---|---|---|---|---|---|---|---|
| MATH500 | $N$ | SC | PoLR | $\eta(\%)$ | $k_t$ (ms) | SC | PoLR | $\eta(\%)$ | $k_t$ (ms) | SC | PoLR | $\eta(\%)$ | $k_t$ (ms) |
| (500) | 51 | 76.2 | -0.8 | 52.4 | 6.5 | 89.8 | -0.4 | 48.7 | 7.6 | 91.8 | 0.2 | 51.8 | 11.2 |
| | 31 | 76.4 | 0.0 | 52.0 | 4.4 | 89.6 | 0.1 | 48.5 | 5.1 | 91.9 | 0.0 | 54.2 | 5.7 |
| | 11 | 75.9 | -1.6 | 52.0 | 2.0 | 89.4 | 0.0 | 48.4 | 2.2 | 91.6 | 0.0 | 60.5 | 2.2 |

| | | DSQ7B | | | | Phi-4-15B | | | | QWQ32B | | | |
|---|---|---|---|---|---|---|---|---|---|---|---|---|---|
| AIME24 | $N$ | SC | PoLR | $\eta(\%)$ | $k_t$ (ms) | SC | PoLR | $\eta(\%)$ | $k_t$ (ms) | SC | PoLR | $\eta(\%)$ | $k_t$ (ms) |
| (30) | 51 | 53.3 | -6.7 | 50.9 | 6.3 | 50.0 | 3.3 | 49.5 | 12.1 | 80.0 | 0.0 | 59.7 | 10.8 |
| | 31 | 53.3 | -3.0 | 51.5 | 3.3 | 49.7 | 0.3 | 51.5 | 5.3 | 78.3 | 0.7 | 61.6 | 6.2 |
| | 11 | 54.3 | -3.3 | 49.8 | 2.0 | 50.3 | -5.3 | 58.6 | 2.2 | 78.3 | -2.0 | 66.6 | 12.8 |

| | | DSQ7B | | | | Phi-4-15B | | | | QWQ32B | | | |
|---|---|---|---|---|---|---|---|---|---|---|---|---|---|
| AIME25 | $N$ | SC | PoLR | $\eta(\%)$ | $k_t$ (ms) | SC | PoLR | $\eta(\%)$ | $k_t$ (ms) | SC | PoLR | $\eta(\%)$ | $k_t$ (ms) |
| (30) | 51 | 33.3 | 0.0 | 48.8 | 7.8 | 33.3 | 0.0 | 54.7 | 11.0 | 76.7 | -10.0 | 56.8 | 12.3 |
| | 31 | 35.3 | 0.0 | 48.4 | 3.9 | 32.0 | 4.0 | 54.8 | 5.9 | 75.0 | -4.0 | 59.5 | 6.9 |
| | 11 | 33.7 | 2.7 | 48.9 | 2.2 | 35.7 | 2.3 | 60.5 | 2.3 | 74.7 | -6.0 | 65.3 | 5.3 |

| | | DSQ7B | | | | MiMo-RL-7B | | | | QWQ32B | | | |
|---|---|---|---|---|---|---|---|---|---|---|---|---|---|
| GPQA DIAMOND | $N$ | SC | PoLR | $\eta(\%)$ | $k_t$ (ms) | SC | PoLR | $\eta(\%)$ | $k_t$ (ms) | SC | PoLR | $\eta(\%)$ | $k_t$ (ms) |
| (198) | 51 | 57.1 | -1.5 | 57.1 | 9.5 | 65.7 | -0.5 | 51.4 | 9.0 | 68.7 | 1.5 | 53.8 | 17.4 |
| | 31 | 55.3 | -1.2 | 55.8 | 5.7 | 64.9 | -1.2 | 51.7 | 5.9 | 68.2 | 0.0 | 56.9 | 7.3 |
| | 11 | 54.1 | -1.7 | 55.4 | 2.4 | 64.6 | 0.0 | 48.9 | 2.5 | 67.9 | 0.0 | 64.3 | 2.5 |

QWQ32B from $91.8\%$ to **92.0%** at $N = 51$, and DSQ7B from $89.6\%$ to **89.7%** at $N = 31$. On AIME25, PoLR boosts accuracy by +3 percentage points for DSQ7B ($33.7\% \rightarrow$ **36.4%**) and Phi-4-15B ($32.0\% \rightarrow$ **36.0%**). These gains occur because PoLR emphasizes the dominant, coherent reasoning clusters while filtering out noisy or inconsistent paths. On few occasions, PoLR downgrades the SC accuracy by small magnitudes except on the AIME25 dataset, where for QWQ32B PoLR is 10 points below SC. This amounts to dropping accuracy on only 3 samples out of total 30 samples in this dataset. In Appendix H, we find that these instances are inherently challenging, with even SC succeeding only by a narrow margin.

Finally, PoLR's benefits are consistent across models, datasets, and trace expansion budgets. On challenging benchmarks like GPQA-DIAMOND, PoLR improves QWQ32B accuracy from $68.7\%$ to **70.2%** at $N = 51$, while reducing token usage nearly by half ($\eta = 53.8\%$). Even on math-intensive datasets such as MATH500, AIME24, and AIME25, PoLR maintains high efficiency and accuracy, demonstrating robustness across reasoning domains and model scales. Additionally, PoLR shows consistent gains over non-math/STEM task, STRATEGYQA, discussed in Appendix G.

In summary, **PoLR halves token usage, preserves or improves accuracy, and adds negligible overhead**, offering a practical, training-free approach to make Self-Consistency substantially more efficient for real-world deployment.

## 5 ANALYSIS AND DISCUSSION

### 5.1 PoLR AS A COMPLEMENT TO ADAPTIVE REASONING.

We evaluate whether *PoLR* can improve adaptive inference methods such as Adaptive Consistency (AC) Aggarwal et al. (2023) and Early-stopping Self-Consistency (ESC) on the GPQA-DIAMOND benchmark. Table 3 reports accuracy and the number of path expansions (*PExp*) across three LLMs and different initial sample budgets $N$.

Table 3: PoLR complements Adaptive Consistency (AC) and Early-Stopping Self-Consistency (ESC) on GPQA-DIAMOND. In the hybrid setting, PoLR ignores low-quality reasoning paths (*PExp*) before adaptive allocation. Results for three LLMs and multiple budgets $N$ show reduced PExp while preserving or improving accuracy. Bold indicates the best result.

| LLMs → | DSQ7B | | | | MiMo-RL-7B | | | | QWQ32B | | | |
|---|---|---|---|---|---|---|---|---|---|---|---|---|
| N → | 51 | | 31 | | 51 | | 31 | | 51 | | 31 | |
| | Acc | *PExp* | Acc | *PExp* | Acc | *PExp* | Acc | *PExp* | Acc | *PExp* | Acc | *PExp* |
| CoT | 54.55 | 1.00 | 54.54 | 1.00 | 63.64 | 1.00 | 63.18 | 1.00 | 68.69 | 1.00 | 68.33 | 1.00 |
| SC | 57.07 | 51.00 | 55.25 | 31.00 | 65.66 | 51.00 | 64.90 | 31.00 | 69.00 | 51.00 | 68.19 | 31.00 |
| PoLR | 55.56 | 20.79 | 54.04 | 13.01 | 65.16 | 24.12 | 63.74 | 14.62 | 70.20 | 21.83 | 67.80 | 12.58 |
| AC | **56.57** | 18.05 | 55.20 | 13.54 | **65.66** | 13.15 | 64.85 | 10.64 | 69.70 | 10.43 | 68.13 | 9.66 |
| PoLR+AC | 55.56 | **10.58** | **55.56** | **10.53** | 65.15 | **9.70** | **65.15** | **9.75** | **70.71** | **8.11** | **70.61** | **8.05** |
| ESC | **56.06** | 24.69 | **55.81** | 18.04 | **65.40** | 19.00 | **64.80** | 14.33 | 68.54 | 16.90 | **67.58** | 13.02 |
| PoLR+ESC | 54.85 | **13.49** | 53.99 | **9.54** | 65.10 | **11.79** | 64.50 | **8.93** | **69.90** | **11.13** | 67.17 | **8.05** |

The results show that PoLR effectively ignores low-quality reasoning paths before applying adaptive methods, reducing the number of expansions required without compromising accuracy. For example, in DSQ7B with $N = 31$, PoLR+AC reduces *PExp* from 13.54 (AC alone) to 10.53, while maintaining comparable accuracy (55.56% vs. 55.20%). Similar patterns hold across MiMo-RL-7B and QWQ32B, with PoLR+AC and PoLR+ESC consistently lowering path expansions by 31.4% (on average) while preserving or slightly improving performance.

These findings indicate that PoLR can serve as an efficient pre-processing step for adaptive reasoning methods. By combining PoLR with adaptive allocation, the hybrid approach achieves substantial computational savings (75% on average as compared to SC) while retaining solution quality, making it a practical enhancement for inference-time reasoning in large language models. All results report mean accuracies over 10 random trials. Table 7 in Appendix D provides standard errors, showing that PoLR yields more precise results with lower variance. We also conducted the same comparison on MATH500, observing consistent patterns (Table 8, Appendix D), confirming the robustness of PoLR across datasets.

## 5.2 LIGHTWEIGHT PREFIX EMBEDDINGS ARE SUFFICIENT FOR POLR.

We evaluate the effect of different prefix embeddings on PoLR performance. Table 4 compares lightweight TF–IDF features with dense semantic embeddings from `tomaarsen/mpnet-base-nli-matryoshka`[2] generating 64-dimensional sentence embedding.

The results show that TF–IDF achieves nearly identical accuracy and token efficiency to dense embeddings while incurring dramatically lower clustering overhead (5–11 ms vs. 220 ms). This indicates that lightweight representations are sufficient for PoLR's prefix clustering: they capture enough structural similarity among reasoning paths. Dense embeddings, while semantically richer, provide little benefit for short prefixes and introduce substantial computational cost. For longer prefixes or highly heterogeneous tasks,

Table 4: Impact of different embeddings on PoLR accuracy on GSM8K, GPQA-Diamond and Math500 datasets.

| | Model | SC | TF-IDF | | | Matryoshka | | |
|---|---|---|---|---|---|---|---|---|
| | | Acc | Acc | $\eta$ | $k_t$ | Acc | $\eta$ | $k_t$ |
| GSM8K | DSQ1.5B | 73.2 | 73.2 | 40.1 | 11.3 | 73.3 | 38.7 | 219.8 |
| | DSQ7B | 79.8 | 80.0 | 26.5 | 5.9 | 79.9 | 26.6 | 219.0 |
| | QWQ32B | 90.8 | 90.8 | 47.6 | 11.2 | 90.8 | 45.8 | 221.4 |
| GPQA DIAMOND | DSQ7B | 57.1 | 55.6 | 57.1 | 9.5 | 54.0 | 52.4 | 209.3 |
| | MIMO7B | 65.7 | 65.2 | 51.4 | 9.0 | 64.1 | 47.7 | 221.5 |
| | QWQ32B | 68.7 | 70.2 | 53.8 | 17.4 | 70.7 | 54.1 | 218.5 |
| MATH500 | DSQ1.5B | 76.2 | 75.4 | 52.4 | 6.5 | 76.2 | 47.5 | 219.7 |
| | DSQ7B | 89.8 | 89.4 | 48.7 | 7.6 | 90.0 | 44.7 | 220.6 |
| | QWQ32B | 91.8 | 92.0 | 51.8 | 11.2 | 91.8 | 51.4 | 218.9 |

denser embeddings may be advantageous, but for the current reasoning benchmarks, lightweight features offer the best trade-off between efficiency and accuracy.

---

[2]https://huggingface.co/tomaarsen/mpnet-base-nli-matryoshka

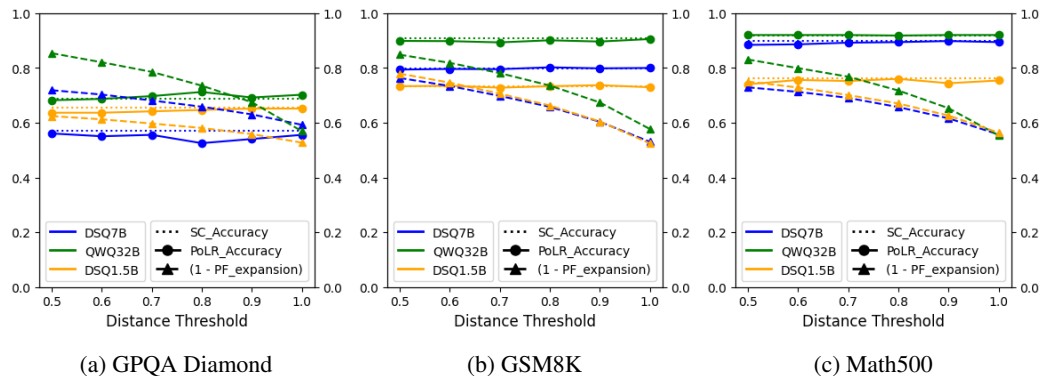

Figure 2: Impact of different cluster threshold selection.

## 5.3 IMPACT OF DISTANCE THRESHOLD IN CLUSTERING.

In PoLR, agglomerative clustering is applied to TF–IDF embeddings of reasoning prefixes, with the distance threshold controlling the granularity of clusters. To study its effect, we vary the threshold from $0.5$ to $1.0$ while fixing the number of samples at $N = 51$, and report accuracy and token efficiency $\eta$ on GPQA-DIAMOND in Figure 2a, GSM8K in Figure 2b and MATH500 in Figure 2c across different LLMs.

Across all thresholds, we find that accuracy remains nearly identical to SC, confirming that **prefix consensus is strong enough that the precise clustering granularity does not affect the final outcome**. However, the **token efficiency is sensitive to the threshold**: lower thresholds lead to tighter clusters, which reduce the size of the dominant cluster and hence the number of traces that need to be expanded. This naturally improves token efficiency.

The magnitude of efficiency gains also depends on model capacity. Weaker models such as DSQ1.5B show the largest improvements (up to $\sim 60\%$ token savings), since they tend to produce more redundant and low-quality traces that can be easily filtered. In contrast, stronger models such as QWQ32B, which generate more diverse yet useful reasoning steps, leave less redundancy to exploit, yielding smaller efficiency gains ($\sim 40\%$). This behavior is consistent with the intuition that PoLR benefits most when the model's reasoning space is noisy and contains many unpromising paths. We choose a threshold of $1.0$ for our main experiments as it **strikes a balance between efficiency and accuracy**: at this setting, the accuracy either matches or slightly exceeds SC, while still providing substantial token savings, whereas lower thresholds could marginally improve efficiency but risk fragmenting clusters unnecessarily.

## 5.4 POLR IS ROBUST TO CLUSTERING METHODS

Table 5 shows that the choice of clustering method has minimal impact on accuracy, but strongly affects PoLR's efficiency metrics. Across GSM8K, MATH500 and GPQA-DIAMOND datasets and different model sizes, density-based methods (DBSCAN and HDBSCAN) yield the better token efficiency ($\eta$), indicating more effective reuse of prefix consensus. These gains come with a modest increase in overhead ($k_t$). Larger models such as QWQ32B benefit most, achieving both high accuracy

Table 5: Impact of different clustering methods on PoLR.

| GSM8K | SC | Agglomerative | | | DBSCAN | | | HDBSCAN | | |
|---|---|---|---|---|---|---|---|---|---|---|
| | Acc | Acc | $\eta$ | $k_t$ | Acc | $\eta$ | $k_t$ | Acc | $\eta$ | $k_t$ |
| DSQ1.5B | 73.2 | 73.2 | 40.1 | 11.3 | 72.7 | 56.1 | 12.0 | 73.5 | 61.5 | 12.9 |
| DSQ7B | 79.8 | 80.0 | 26.5 | 05.9 | 80.1 | 36.4 | 06.9 | 80.4 | 33.4 | 07.1 |
| QWQ32B | 90.8 | 90.8 | 47.6 | 11.2 | 90.8 | 64.6 | 11.4 | 91.0 | 56.9 | 12.0 |

| GPQA | SC | Agglomerative | | | DBSCAN | | | HDBSCAN | | |
|---|---|---|---|---|---|---|---|---|---|---|
| | Acc | Acc | $\eta$ | $k_t$ | Acc | $\eta$ | $k_t$ | Acc | $\eta$ | $k_t$ |
| DSQ7B | 57.1 | 55.6 | 57.1 | 09.5 | 56.1 | 69.9 | 10.2 | 53.5 | 71.8 | 11.0 |
| MIMO7B | 65.7 | 65.2 | 51.4 | 09.0 | 64.1 | 65.3 | 10.1 | 63.6 | 67.0 | 10.5 |
| QWQ32B | 68.7 | 70.2 | 53.8 | 17.4 | 68.2 | 52.2 | 14.9 | 68.2 | 55.2 | 15.3 |

| Math500 | SC | Agglomerative | | | DBSCAN | | | HDBSCAN | | |
|---|---|---|---|---|---|---|---|---|---|---|
| | Acc | Acc | $\eta$ | $k_t$ | Acc | $\eta$ | $k_t$ | Acc | $\eta$ | $k_t$ |
| DSQ1.5B | 76.2 | 75.4 | 52.4 | 6.5 | 76.2 | 68.7 | 7.2 | 76.0 | 67.4 | 7.7 |
| DSQ7B | 89.8 | 89.4 | 48.7 | 7.6 | 89.6 | 54.6 | 8.2 | 89.2 | 63.5 | 9.7 |
| QWQ32B | 91.8 | 92.0 | 51.8 | 11.2 | 92.0 | 71.3 | 12.6 | 92.2 | 63.1 | 12.6 |

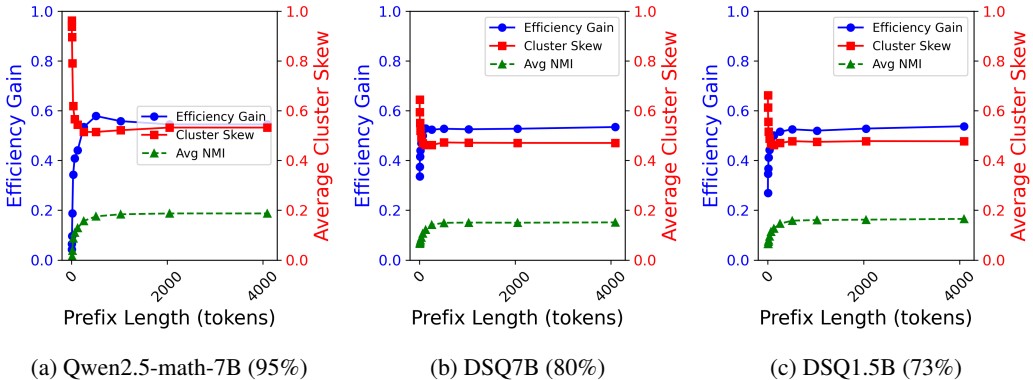

(a) Qwen2.5-math-7B (95%)  (b) DSQ7B (80%)  (c) DSQ1.5B (73%)

Figure 3: Efficiency gains of PoLR across three models on GSM8K as a function of prefix length. All models achieve over 50% token savings by 256-512 prefix tokens.

and strong efficiency improvements. Overall, we observed that the **clustering methods mainly impact the efficiency–overhead trade-off, not the accuracy, showing that the PoLR is robust to different clustering methods.**

### 5.5 Effect of Prefix Length on Efficiency and Cluster Structure.

To further understand the role of prefix information, we varied prefix length from 2 to 4096 tokens on GSM8K with different LLMs. Figure 3 shows the resulting efficiency gains, cluster skew, and NMI. Raw numbers are provided in Appendix E, Table 9.

We observe that efficiency improves monotonically with prefix length up to $\sim 512$ tokens, achieving $\sim 58\%$ token savings over Self-Consistency, after which it saturates. Cluster skew, by contrast, decreases sharply from $\sim 0.96$ at length 2 to $\sim 0.52$ at 256, stabilizing thereafter. NMI increases slowly with prefix length but remains relatively low ($\sim 0.18$ at 4096).

These results highlight two insights that efficiency is primarily governed by structural dominance (skew) rather than correctness alignment (NMI). Even with weak NMI, PoLR achieves substantial token savings whenever a dominant cluster exists. Second, prefix length trades off skew and NMI. Short prefixes yield high skew but low predictiveness; longer prefixes reduce skew while slightly improving correctness alignment. PoLR's efficiency benefits emerge in the mid-range, when skew remains sufficient for pruning yet prefixes capture more reasoning structure.

We compared PoLR across three LLMs for GSM8K dataset with differing accuracies (73%–95%). Across all models, efficiency gains eventually plateau around 50–55%, but the trajectory differs. Lower-capacity models achieve large savings even at very short prefixes (2–16 tokens), whereas the higher-accuracy Qwen2.5-Math-7B requires longer prefixes (256–512) before efficiency saturates. Importantly, cluster skew consistently predicts efficiency gains, while NMI remains low across all models. This highlights that PoLR's benefits stem from structural dominance of prefix clusters rather than their correctness alignment, and that model capacity mainly shifts the prefix length required to realize these savings.

### 5.6 Cluster size $\implies$ Better Accuracy

To further understand which cluster's reasoning traces should be expanded, we perform the majority voting for all the formed clusters for all dataset model combinations. We observed that the dominant cluster (by cluster size) captures more accurate traces while the remaining clusters tend to have lower accuracies compared to the dominant cluster, second dominant cluster being typically ranking second in accuracy, and so on.

Table 6: Correlation coefficient between cluster sizes and accuracy for DSQ7B and QwQ32B ($N = 51, L_p = 256$).

|  | GSM8K | MATH500 | AIME24 | AIME25 | GPQA |
|---|---|---|---|---|---|
| DSQ-7B | 0.90 | 0.89 | 0.75 | 0.76 | 0.90 |
| QwQ-32B | 0.90 | 0.88 | 0.87 | 0.78 | 0.78 |

We further find the Pearson correlation coefficient $\rho$ between the cluster sizes and the corresponding accuracies for all the dataset in Table 6. We observe a strong correlation for all dataset model combinations $\rho > 0.75$. Therefore, in PoLR, **cluster size is the best indicator of the cluster to be expanded**. We provide the prefix length-wise dominant cluster accuracy for all dataset model combinations in Figure 5 Appendix F.

We refer the reader to Appendix I for further analysis of the impact of other hyperparameters on PoLR.

## 6 RELATED WORK

We focus on two main directions relevant to our work: (1) methods that utilize answer consistency to reduce inference cost, and (2) methods that exploit early reasoning prefixes.

**Methods Exploiting Answer Consistency**  Self-Consistency (SC) Wang et al. (2023) has become a standard approach for improving the reliability of chain-of-thought reasoning by sampling multiple solution paths and selecting the majority answer. Other verifier-based methods include Cobbe et al. (2021); Uesato et al. (2022); Yao et al. (2023). While effective, SC is inefficient: accuracy improves roughly linearly with the number of samples $N$, but decoding cost scales proportionally, leading to substantial redundancy when many trajectories repeat similar reasoning patterns.

Several methods aim to mitigate this overhead. Adaptive Consistency (AC) Aggarwal et al. (2023) monitors answer agreement as samples arrive, allocating fewer trajectories to "easy" problems where consensus forms quickly. Early-Stopping Self-Consistency (ESC) Li et al. (2024) halts sampling once a confident majority is detected, avoiding the cost of decoding all $N$ samples. Reasoning-Aware Self-Consistency (RASC) Wan et al. (2025) evaluates reasoning paths based on a set of features and aggregates answers using weighted majority voting after collecting high-quality paths.

Both AC and ESC reduce compute by relying on answer-level agreement, but they act *after* full trajectories are decoded. In contrast, **PoLR exploits structural signals much earlier: by clustering prefixes before expansion, it avoids generating redundant modes upfront rather than waiting for consensus at the answer level**. This is crucial because when agreement is delayed or split across modes, AC and ESC still expend tokens unnecessarily, whereas PoLR prevents this overhead entirely. Our experiments show that PoLR and AC are complementary: prefix clustering removes redundant modes, while adaptive stopping controls allocation within the dominant cluster, achieving the strongest efficiency–accuracy trade-offs. Moreover, unlike iterative methods, **PoLR is highly parallelizable**, leading to higher throughput in practice since multiple promising prefixes can be expanded simultaneously without waiting for sequential majority-vote checks.

**Methods Exploiting Early Prefixes**  Another line of work leverages the predictive power of early prefixes. Path Consistency Zhu et al. (2024) estimates the confidence of partial reasoning paths and guides subsequent generations toward promising branches. In contrast, **PoLR does not rely on external confidence estimators or guided decoding**. Instead, it clusters multiple short prefixes to capture naturally emerging consensus among independent samples and applies self-consistency voting only within that cluster. This preserves SC's majority-vote principle while substantially reducing computational cost.

Similarly, UPFT Ji et al. (2025) shows that prefixes contain rich signals and uses them at *training time* for supervision. **PoLR transfers this insight to inference**, demonstrating that prefixes can be exploited *unsupervised and training-free* to reduce inference cost while maintaining SC-level accuracy. Orthogonal to prefix-based methods, several trainable approaches iteratively leverage LLM outputs to improve model performance Zelikman et al. (2022); Yuan et al. (2023).

## 7 CONCLUSION

In this work, we present PoLR, which leverages prefix clustering to drastically reduce reasoning cost while preserving or improving accuracy, providing a training-free, inference-time enhancement to Self-Consistency.

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

## A  HYPERPARAMETER SETTINGS

In this section we provide the hyperparameter settings for PoLR and the other comparative methods. All experiments were conducted using LightevalHabib et al. (2023), supporting 7,000+ evaluation tasks across multiple domains and languages. We performed all experiments on 4 NVIDIA L40S 48GB memory cards. We now define the core parameters for each method used in this work. For the comparative methods we used the hyperparameters configurations yielding the best performance.

---

**Algorithm 1** Path of Least Resistance (PoLR)

---

**Require:** Question $x$, LLM $\mathcal{M}$, prefix length $L_p$, #prefixes $N$, #expansions $K$
**Ensure:** Final answer $\hat{a}$
1: **Prefix Sampling:**
2: **for** $i = 1 \ldots N$ **do**
3: $\quad p_i \leftarrow \text{Prefix}(\mathcal{M}(x, t_i), L_p)$
4: **end for**
5: **Clustering:**
6: Embed prefixes: $e_i \leftarrow \text{Embed}(p_i)$
7: $\mathcal{C} \leftarrow \text{Cluster}(\{e_i\}_{i=1}^{N})$
8: $C^* \leftarrow \arg\max_{C_j \in \mathcal{C}} |C_j|$
9: **Expansion:**
10: Select $K$ prefixes $\{p_1, \ldots, p_K\} \subset C^*$
11: **for** $k = 1 \ldots K$ **do**
12: $\quad r_k \leftarrow \mathcal{M}(x \mid p_k)$ $\qquad\qquad\qquad\qquad\qquad$ ▷ Complete reasoning trace
13: $\quad a_k \leftarrow \text{ExtractAnswer}(r_k)$
14: **end for**
15: **Self-Consistency Voting:**
16: $\hat{a} \leftarrow \arg\max_y \sum_{k=1}^{K} \mathbf{1}[a_k = y]$
17: **return** $\hat{a}$

---

**PoLR**

- top-p=0.9,
- temperature=0.6,
- max-token=32K,
- prefix-Length=256,
- clustering parameters:
  - clustering distance threshold=1.0
  - feature downsampling dim=10
  - linkage=average,
  - metric=cosine,

**Adaptive Consistency (AC)**   Aggarwal et al. (2023)

- top-p=0.9,
- temperature=0.6,
- max-token=32K,
- stop criteria: $\beta-$confidence threshold=0.95,

**Early-stopping Self-consistency (ESC)**   Li et al. (2024)

- top-p=0.9,
- temperature=0.6,
- max-token=32K,
- window-size=5,

## A.1   LLM Usage

We used ChatGPT and Claude for grammar reviews and language style polishing. In certain cases we used these models in analyzing and summarizing tables. These summaries are then verified and updated manually for correctness.

## B  PoLR ALGORITHM

Algorithm 1 provide the step-by-step instruction to implement PoLR. PoLR is model agnostic and wokrs for any LLM.

## C  PoLR PERFORMANCE VS PREFIX LENGTHS

Following the structure of Table 2, we report mean accuracies and standard deviations for both PoLR and SC across different prefix lengths $L_p \in \{1, 2, 4, 8, 16, 32, 64, 128, 256, 512, 1024, 2048, 4096\}$ and two sampling budgets $N \in \{51, 31\}$ (Figure 4).

Overall, PoLR exhibits remarkable robustness to the initial number of samples: for both $N = 51$ and $N = 31$, token efficiency follows nearly identical curves. In both settings, we observe a sharp improvement in efficiency once prefix lengths reach the range 128–512, after which efficiency plateaus or declines slightly. The drop at very long prefixes arises because many instances do not require extended prefixes to reach a stable answer—expanding them wastes computation without improving consensus. This trend is consistent across all dataset–LLM pairs we tested.

In terms of quality, PoLR generally matches or outperforms SC across prefix lengths. The gains are most stable on math and commonsense datasets (e.g., GSM8K, MATH500, AIME24/25), where prefix consensus is especially strong. The only exception is GPQA-DIAMOND, where accuracy drops slightly for longer prefixes. We attribute this to the nature of GPQA problems: they often require multi-step reasoning and longer prefixes often contains specialized technical terms that leads to less informative lexical overlap between prefixes. Potential solutions could include expanding top-$m$ clusters instead of the dominant cluster or switching to semantic neural embeddings. We leave this for future work.

## D  PoLR COMPARISON WITH EXISTING APPROACHES

### D.1  PoLR COMPLEMENTS ADAPTIVE AND EARLY-STOPPING CONSISTENCY ACROSS DATASETS

Tables 7 and 8 evaluate PoLR in combination with Adaptive Consistency (AC) and Early-Stopping Consistency (ESC) on two contrasting benchmarks: GPQA-DIAMOND (STEM reasoning with highly diverse, less predictable traces) and MATH500 (structured math reasoning with strong prefix regularities).

**GPQA-DIAMOND.**  On GPQA, prefixes are less predictive of correctness, making consensus weaker. Here, PoLR alone already reduces expansions substantially (e.g., DSQ7B: 20.79 vs. 51 under SC at $N = 51$), but occasionally trails SC in accuracy. However, when combined with AC or ESC, PoLR consistently lowers *PExp* by an additional 30–40% while preserving or even slightly improving accuracy. For instance, PoLR+AC with DSQ7B cuts expansions to 10.58 from 18.05 under AC, with no loss in performance. This highlights PoLR's role as a *pruning front-end* that removes clearly redundant reasoning paths before adaptive allocation.

**MATH500.**  On structured math problems, prefix clusters are highly reliable. PoLR alone reduces expansions by more than half (e.g., DSQ7B: 22.47 vs. 51 at $N = 51$) while matching SC accuracy. When combined with AC, expansions drop to as low as 5.69 per problem (DSQ7B, $N = 51$) without measurable accuracy loss, achieving up to $5\times$ efficiency gains. ESC also benefits: PoLR+ESC consistently reduces expansions (e.g., 9.23 vs. 10.66 on QWQ32B) while retaining SC-level performance.

**Takeaway.**  The contrast between GPQA-DIAMOND and MATH500 illustrates the conditions under which PoLR is most effective. On tasks with highly structured reasoning (MATH500), PoLR is nearly lossless and compounds the efficiency of adaptive strategies to yield massive savings. On tasks with more diverse or less predictable reasoning paths (GPQA), PoLR still reduces redundancy and enhances existing adaptive methods, though accuracy gains are less pronounced. Together, these

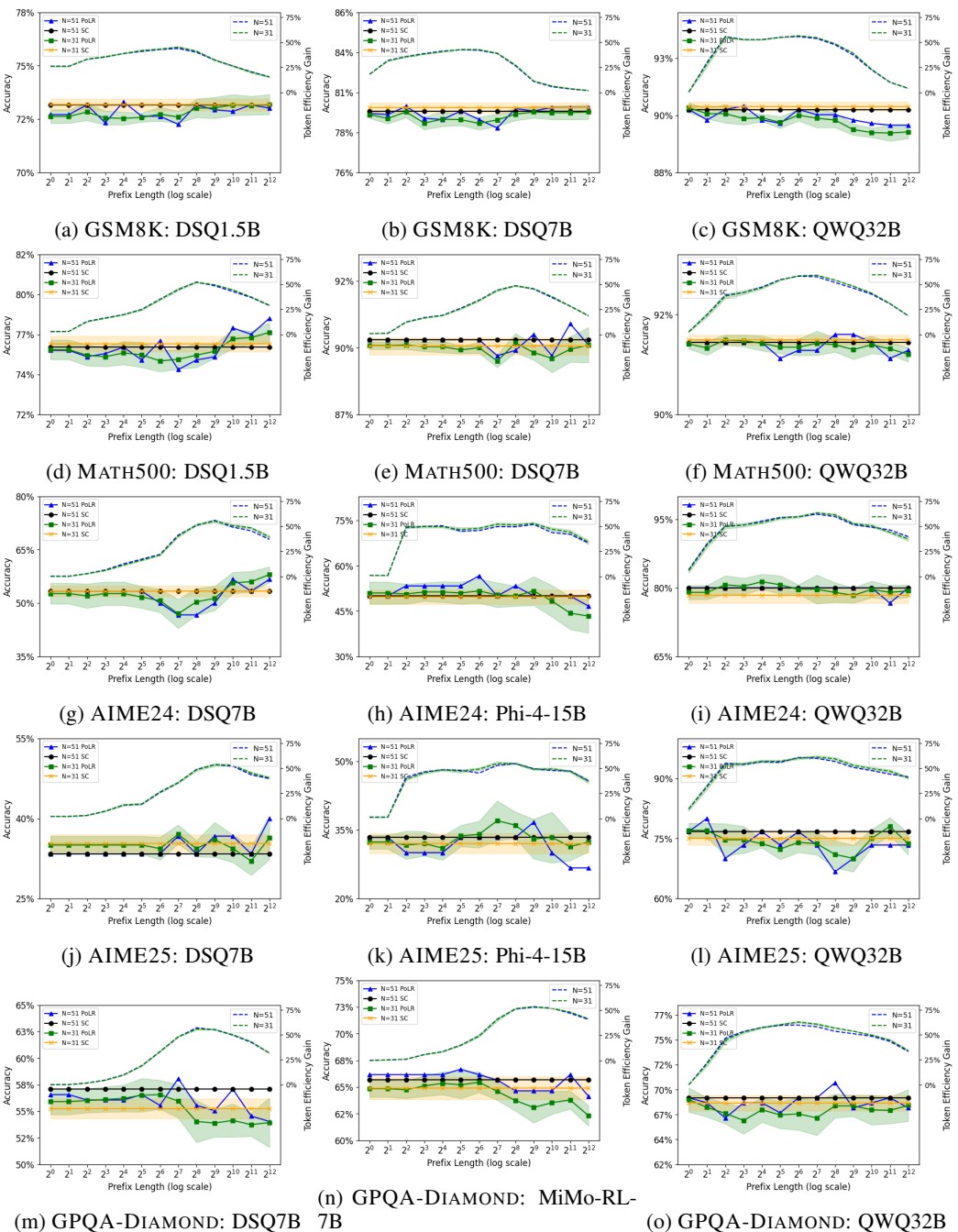

(a) GSM8K: DSQ1.5B     (b) GSM8K: DSQ7B     (c) GSM8K: QWQ32B

(d) MATH500: DSQ1.5B     (e) MATH500: DSQ7B     (f) MATH500: QWQ32B

(g) AIME24: DSQ7B     (h) AIME24: Phi-4-15B     (i) AIME24: QWQ32B

(j) AIME25: DSQ7B     (k) AIME25: Phi-4-15B     (l) AIME25: QWQ32B

(n) GPQA-DIAMOND: MiMo-RL-

(m) GPQA-DIAMOND: DSQ7B     7B     (o) GPQA-DIAMOND: QWQ32B

Figure 4: Performance comparison of PoLR versus SC across datasets (GSM8K, MATH500, AIME24, AIME25, GPQA-DIAMOND) and model sizes. The table shows accuracy differences (green = improvement, red = drop), token efficiency $\eta$ (%), and sample size $N$ as a function of different prefix lengths $L_p$.

Table 7: PoLR complements Adaptive Consistency (AC) and Early-Stopping Consistency (ESC) on GPQA-DIAMOND. In the hybrid setting, PoLR prunes low-quality reasoning paths (*PExp*) before adaptive allocation. Results for three LLMs and multiple budgets $N$ show reduced PExp while preserving or improving accuracy.

(a) DSQ7B

| LLMs → | DSQ7B | | | |
|---|---|---|---|---|
| N → | 51 | | 31 | |
| | Acc | *PExp* | Acc | *PExp* |
| CoT | $54.54 \pm 0.00$ | $1.00 \pm 0.00$ | $54.54 \pm 0.00$ | $1.00 \pm 0.00$ |
| SC | $57.07 \pm 0.00$ | $51.00 \pm 0.00$ | $55.25 \pm 0.93$ | $31.00 \pm 0.00$ |
| PoLR | $55.55 \pm 0.00$ | $20.79 \pm 0.00$ | $54.04 \pm 1.96$ | $13.01 \pm 0.15$ |
| AC | $56.57 \pm 0.00$ | $18.05 \pm 0.00$ | $55.20 \pm 0.90$ | $13.54 \pm 0.34$ |
| PoLR + AC | $55.56 \pm 0.00$ | $10.58 \pm 0.00$ | $55.56 \pm 0.00$ | $10.53 \pm 0.06$ |
| ESC | $56.06 \pm 0.78$ | $24.69 \pm 0.87$ | $55.80 \pm 1.51$ | $18.04 \pm 0.63$ |
| PoLR+ESC | $54.84 \pm 0.40$ | $13.48 \pm 0.36$ | $53.99 \pm 1.72$ | $9.53 \pm 0.25$ |

(b) MiMo-RL-7B

| LLMs → | MiMo-RL-7B | | | |
|---|---|---|---|---|
| N → | 51 | | 31 | |
| | Acc | *PExp* | Acc | *PExp* |
| CoT | $63.63 \pm 0.00$ | $1.00 \pm 0.00$ | $63.18 \pm 0.00$ | $1.00 \pm 0.00$ |
| SC | $65.65 \pm 0.00$ | $51.00 \pm 0.00$ | $64.89 \pm 1.08$ | $31.00 \pm 0.00$ |
| PoLR | $65.15 \pm 0.00$ | $24.11 \pm 0.00$ | $63.73 \pm 0.97$ | $14.62 \pm 0.14$ |
| AC | $65.66 \pm 0.00$ | $13.15 \pm 0.00$ | $64.84 \pm 1.11$ | $10.64 \pm 0.26$ |
| PoLR + AC | $65.15 \pm 0.00$ | $9.70 \pm 0.00$ | $65.15 \pm 0.00$ | $9.75 \pm 0.06$ |
| ESC | $65.40 \pm 0.60$ | $19.01 \pm 0.98$ | $64.79 \pm 0.93$ | $14.32 \pm 0.57$ |
| PoLR+ESC | $65.10 \pm 0.35$ | $11.78 \pm 0.45$ | $64.49 \pm 0.84$ | $8.93 \pm 0.15$ |

(c) QWQ32B

| LLMs → | QWQ32B | | | |
|---|---|---|---|---|
| N → | 51 | | 31 | |
| | Acc | *PExp* | Acc | *PExp* |
| CoT | $68.68 \pm 0.00$ | $1.00 \pm 0.00$ | $68.33 \pm 0.00$ | $1.00 \pm 0.00$ |
| SC | $69.00 \pm 0.00$ | $51.00 \pm 0.00$ | $68.18 \pm 0.78$ | $31.00 \pm 0.00$ |
| PoLR | $70.20 \pm 0.00$ | $21.83 \pm 0.00$ | $67.87 \pm 1.19$ | $12.57 \pm 0.19$ |
| AC | $69.70 \pm 0.00$ | $10.43 \pm 0.00$ | $68.13 \pm 1.02$ | $9.65 \pm 0.33$ |
| PoLR + AC | $70.71 \pm 0.00$ | $8.11 \pm 0.00$ | $70.61 \pm 0.20$ | $8.05 \pm 0.05$ |
| ESC | $68.53 \pm 0.71$ | $16.90 \pm 0.67$ | $67.57 \pm 0.95$ | $13.02 \pm 0.43$ |
| PoLR+ESC | $69.89 \pm 0.68$ | $11.13 \pm 0.24$ | $67.17 \pm 0.87$ | $8.05 \pm 0.12$ |

Table 8: PoLR complements Adaptive Consistency (AC) and Early-Stopping Consistency (ESC) on MATH500. In the hybrid setting, PoLR prunes low-quality reasoning paths (*PExp*) before adaptive allocation. Results for three LLMs and multiple budgets $N$ show reduced PExp while preserving or improving accuracy.

(a) DSQ1.5B

| LLMs → | DSQ1.5B | | | | |
|---|---|---|---|---|---|
| N → | 51 | | | 31 | |
| | Acc | *PExp* | | Acc | *PExp* |
| CoT | $73.00 \pm 0.00$ | $1.00 \pm 0.00$ | | $72.88 \pm 0.00$ | $1.00 \pm 0.00$ |
| SC | $76.20 \pm 0.00$ | $51.00 \pm 0.00$ | | $76.40 \pm 0.49$ | $31.00 \pm 0.00$ |
| PoLR | $75.40 \pm 0.00$ | $22.13 \pm 0.00$ | | $75.70 \pm 0.77$ | $13.39 \pm 0.14$ |
| AC | $78.60 \pm 0.00$ | $12.20 \pm 0.00$ | | $77.84 \pm 0.45$ | $10.19 \pm 0.16$ |
| PoLR + AC | $76.20 \pm 0.00$ | $8.63 \pm 0.00$ | | $76.14 \pm 0.09$ | $8.67 \pm 0.04$ |
| ESC | $76.26 \pm 0.09$ | $27.42 \pm 0.58$ | | $76.22 \pm 0.51$ | $19.69 \pm 0.24$ |
| PoLR+ESC | $75.66 \pm 0.23$ | $15.83 \pm 0.26$ | | $76.12 \pm 0.60$ | $11.26 \pm 0.15$ |

(b) DSQ7B

| LLMs → | DSQ7B | | | | |
|---|---|---|---|---|---|
| N → | 51 | | | 31 | |
| | Acc | *PExp* | | Acc | *PExp* |
| CoT | $89.20 \pm 0.00$ | $1.00 \pm 0.00$ | | $89.12 \pm 0.00$ | $1.00 \pm 0.00$ |
| SC | $89.80 \pm 0.00$ | $51.00 \pm 0.00$ | | $89.56 \pm 0.32$ | $31.00 \pm 0.00$ |
| PoLR | $89.40 \pm 0.00$ | $22.47 \pm 0.00$ | | $89.68 \pm 0.36$ | $13.71 \pm 0.10$ |
| AC | $90.00 \pm 0.00$ | $7.07 \pm 0.00$ | | $89.94 \pm 0.28$ | $6.24 \pm 0.08$ |
| PoLR + AC | $89.40 \pm 0.00$ | $5.69 \pm 0.00$ | | $89.48 \pm 0.10$ | $5.61 \pm 0.04$ |
| ESC | $89.82 \pm 0.06$ | $14.64 \pm 0.24$ | | $89.62 \pm 0.24$ | $11.84 \pm 0.19$ |
| PoLR+ESC | $89.40 \pm 0.00$ | $10.77 \pm 0.11$ | | $89.68 \pm 0.45$ | $9.25 \pm 0.06$ |

(c) QWQ32B

| LLMs → | QWQ32B | | | | |
|---|---|---|---|---|---|
| N → | 51 | | | 31 | |
| | Acc | *PExp* | | Acc | *PExp* |
| CoT | $92.00 \pm 0.00$ | $1.00 \pm 0.00$ | | $91.94 \pm 0.00$ | $1.00 \pm 0.00$ |
| SC | $91.80 \pm 0.00$ | $51.00 \pm 0.00$ | | $91.86 \pm 0.13$ | $31.00 \pm 0.00$ |
| PoLR | $92.00 \pm 0.00$ | $22.78 \pm 0.00$ | | $91.74 \pm 0.18$ | $13.24 \pm 0.08$ |
| AC | $92.20 \pm 0.00$ | $4.87 \pm 0.00$ | | $91.74 \pm 0.18$ | $4.72 \pm 0.06$ |
| PoLR + AC | $92.00 \pm 0.00$ | $4.67 \pm 0.00$ | | $92.00 \pm 0.00$ | $4.65 \pm 0.01$ |
| ESC | $91.79 \pm 0.00$ | $10.66 \pm 0.12$ | | $91.88 \pm 0.15$ | $9.62 \pm 0.08$ |
| PoLR+ESC | $92.00 \pm 0.00$ | $9.23 \pm 0.05$ | | $91.92 \pm 0.24$ | $8.42 \pm 0.03$ |

Table 9: Efficiency gains of PoLR across three models on GSM8K as a function of prefix length. All models achieve over 50% token savings by 256-512 prefix tokens. Here *PEff* denotes $1 - \frac{PExp}{N}$.

| $L_p$ | DSQ7B (80%) | | | DSQ1.5B (73%) | | | Qwen2.5-math-7B (95%) | | |
|---|---|---|---|---|---|---|---|---|---|
| | *PEff* | avg_skew | avg_nmi | *PEff* | avg_skew | avg_nmi | *PEff* | avg_skew | avg_nmi |
| 2 | 0.336 | 0.644 | 0.066 | 0.270 | 0.662 | 0.067 | 0.043 | 0.963 | 0.009 |
| 4 | 0.375 | 0.594 | 0.070 | 0.346 | 0.614 | 0.066 | 0.065 | 0.937 | 0.012 |
| 8 | 0.415 | 0.551 | 0.074 | 0.368 | 0.556 | 0.075 | 0.097 | 0.896 | 0.016 |
| 16 | 0.440 | 0.520 | 0.080 | 0.412 | 0.516 | 0.081 | 0.187 | 0.791 | 0.038 |
| 32 | 0.469 | 0.494 | 0.091 | 0.443 | 0.488 | 0.096 | 0.343 | 0.619 | 0.087 |
| 64 | 0.500 | 0.469 | 0.107 | 0.475 | 0.468 | 0.114 | 0.409 | 0.567 | 0.111 |
| 128 | 0.529 | 0.462 | 0.123 | 0.501 | 0.462 | 0.127 | 0.442 | 0.545 | 0.129 |
| 256 | 0.524 | 0.463 | 0.142 | 0.516 | 0.471 | 0.146 | 0.537 | 0.514 | 0.157 |
| 512 | 0.528 | 0.473 | 0.150 | 0.525 | 0.478 | 0.158 | 0.579 | 0.515 | 0.176 |
| 1024 | 0.526 | 0.471 | 0.151 | 0.520 | 0.475 | 0.161 | 0.558 | 0.522 | 0.184 |
| 2048 | 0.528 | 0.471 | 0.150 | 0.530 | 0.478 | 0.162 | 0.548 | 0.532 | 0.187 |
| 4096 | 0.534 | 0.470 | 0.151 | 0.537 | 0.477 | 0.165 | 0.545 | 0.532 | 0.187 |

Table 10: Efficiency gains of PoLR on STRATEGYQA DSQ7B combination as compared to SC.

| $L_p \rightarrow$ | 2 | 4 | 8 | 16 | 32 | 64 | 128 | 256 | 512 | 1024 | 2048 | 4096 |
|---|---|---|---|---|---|---|---|---|---|---|---|---|
| SC (%) | 59.8 | 59.8 | 59.8 | 59.8 | 59.8 | 59.8 | 59.8 | 59.8 | 59.8 | 59.8 | 59.8 | 59.8 |
| PoLR (%) | 60.0 | 59.0 | 59.0 | 59.8 | 57.9 | 57.9 | 57.6 | 59.6 | 59.8 | 61.1 | 60.5 | 61.8 |
| *PEff* | 0.185 | 0.275 | 0.313 | 0.361 | 0.470 | 0.565 | 0.646 | 0.662 | 0.657 | 0.655 | 0.651 | 0.656 |
| $\eta$ | 0.110 | 0.107 | 0.146 | 0.192 | 0.290 | 0.383 | 0.430 | 0.315 | 0.150 | 0.063 | 0.058 | 0.061 |
| $k_t$ | 1.3 | 1.4 | 1.5 | 1.7 | 3.7 | 7.4 | 11.2 | 9.4 | 12.2 | 12.1 | 13.0 | 13.1 |

results show that PoLR is both a strong standalone alternative to SC and a *universal enhancer* for adaptive consistency methods across reasoning domains.

## E  EFFECT OF PREFIX LENGTH ON EFFICIENCY AND CLUSTER STRUCTURE

In this section, we provide the cluster skew, NMI and efficiencies gains with varying prefix lengths for GSM8K dataset. The plots are provided in Section 5.5 in Figure 3. For reproducibility, we are providing the raw numbers in Table 9 for each of the plots in Figure 3.

## F  CLUSTER SIZE IS THE INDICATOR OF BEST PERFORMANCE

Following the structure of Table 2, we report the cluster-wise self-consistency across different prefix lengths $L_p \in \{1, 2, 4, 8, 16, 32, 64, 128, 256, 512, 1024, 2048, 4096\}$, *cluster 0* being the dominant cluster with highest number of reasoning prefixes, in Figure 5. We observe that dominant cluster consistently delivers best accuracies across the model dataset prefix length combinations except for the AIME24/25 dataset. It is likely because these benchmarks (e.g., AIME with only 30 samples) are relatively small, reducing statistical robustness. Therefore, the plots for AIME seems to be a bit noisy. However, for all the combinations we observed strong correlation between the cluster sizes and the corresponding accuracies for all the dataset $\rho > 0.75$. Therefore, in PoLR, **clustering cardinality is the best indicator of the cluster to be expanded.**

## G  POLR EVALUATION ON NON-MATH DATASETS

To further evaluate the generality of our observations beyond math/STEM settings, we also experimented with a non-math/STEM QA dataset, STRATEGYQA with DSQ7B model. We find that PoLR continues to match SC accuracy preserving the token efficiency benefits as depicted in Table 10.

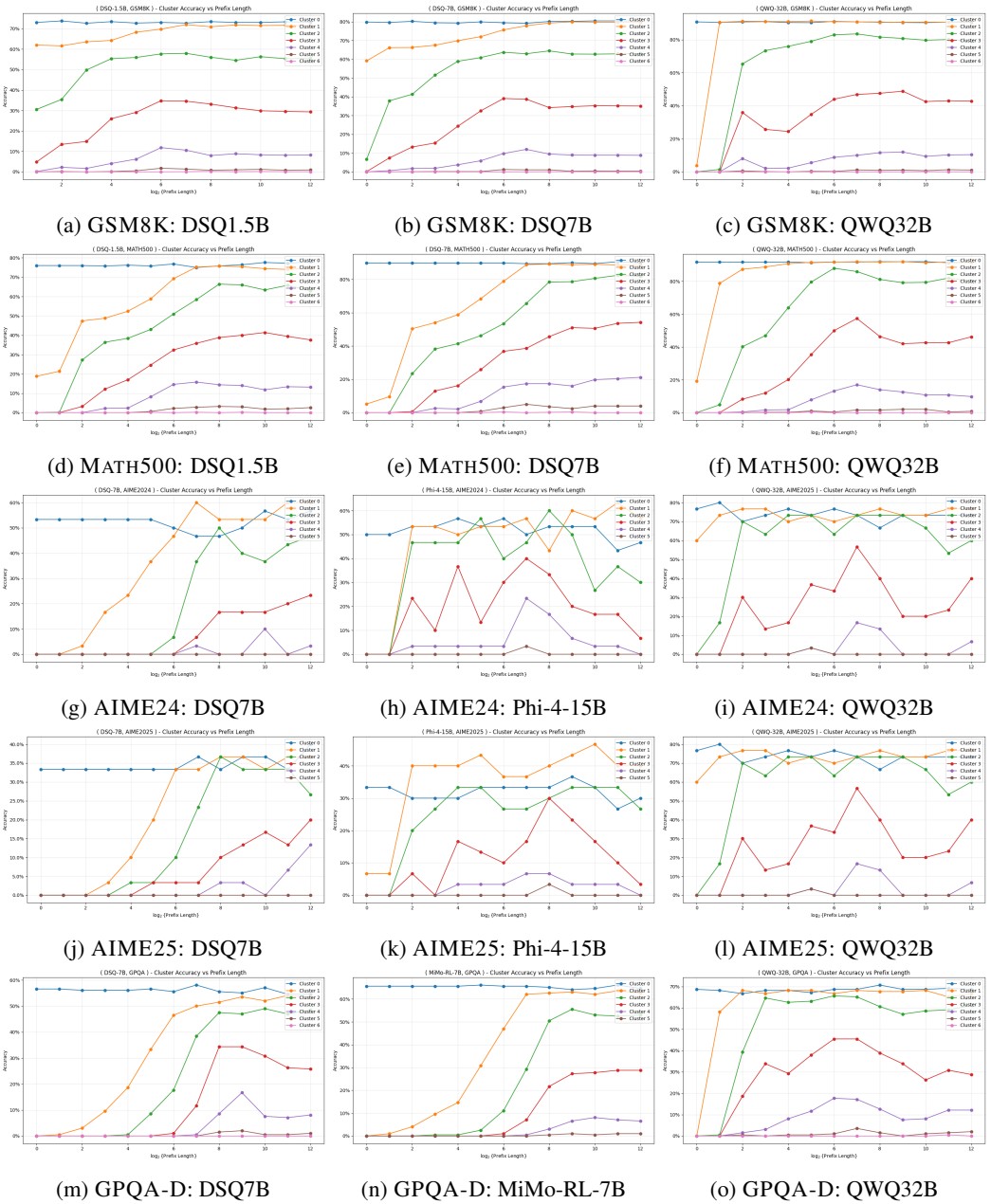

Figure 5: Cluster-wise self-consistency for (GSM8K, MATH500, AIME24, AIME25, GPQA-DIAMOND) datasets with different model sizes.

Table 11: Instance-level error analysis of PoLR on AIME25 dataset QWQ32B model combination.

| | Cluster number | Cluster size | Correct | Incorrect | % Correct Within | SC consensus % |
|---|---|---|---|---|---|---|
| Failure case 1 | 0 | 20 | 9 | 11 | 45.0% | |
| | 1 | 19 | 10 | 9 | 53.0% | 53.0% |
| | 2 | 12 | 8 | 4 | 67.0% | |
| Failure case 2 | 0 | 22 | 9 | 13 | 41.0% | |
| | 1 | 15 | 11 | 4 | 73.0% | 59.0% |
| | 2 | 14 | 10 | 4 | 71.0% | |
| Failure case 3 | 0 | 20 | 9 | 11 | 45.0% | |
| | 1 | 13 | 8 | 5 | 62.0% | |
| | 2 | 13 | 11 | 2 | 85.0% | 63.0% |
| | 3 | 5 | 4 | 1 | 80.0% | |
| Random correct case | 0 | 27 | 26 | 1 | 96.0% | |
| | 1 | 19 | 19 | 0 | 100.0% | 96.0% |
| | 2 | 5 | 4 | 1 | 80.0% | |
| Average case | 0 | 21.77 | 14.6 | 7.2 | 66.1% | |
| | 1 | 14.63 | 10.1 | 4.6 | 68.5% | 67.9% |
| | 2 | 9.70 | 7.0 | 2.7 | 65.3% | |

Across prefix lengths, PoLR maintains accuracy comparable to or slightly better than SC, while achieving strong token efficiency improvements ($\eta$), consistent with our observations on other math and STEM datasets.

## H PoLR Instance-level error analysis

To better understand the observed 10% accuracy drop for QwQ32B on AIME2025 dataset, we perform an instance-level analysis of the cases where SC passes but PoLR fails. Though this drop seems large, AIME2025 contains only 30 samples, meaning a 10% drop corresponds to a deviation of just three problems. Given this small sample size, even a few challenging instances can produce noticeable fluctuations.

In this analysis, for each incorrect instance, we inspect the cluster structures formed by PoLR. For each failure instance, We compute the number of correct reasoning traces within each cluster and compared it against the SC consensus in Table 11. We observe that the SC consensus for these failure cases is substantial low as compared to the average case (last row block in Table 11) indicating that these instances are inherently difficult, even for the SC baseline. Further, all these failure case shows weaker cluster purity making PoLR's selection task more ambiguous. However, the primary contribution of PoLR remains to be the substantial improvements in token efficiency and cost reduction, rather than surpassing SC in raw accuracy. Therefore, **for the challenging problems where even the SC baseline solves the task only narrowly, PoLR is not expected to outperform SC in accuracy.**

## I Impact of Temperature Sampling

To evaluate whether dynamically adjusting the prefix length based on the sampling temperature can improve performance, we conduct a controlled study on MATH500 using DSQ7B. We vary prefix lengths from 1 to 4096 and LLM sampling temperatures from 0.2 to 1.0 in Table 12.

Across all temperatures and prefix lengths, the accuracy varies only within a narrow band of $\pm 1$ points, showing no consistent trend that correlates temperature with an optimal prefix length suggesting that sampling temperature does not meaningfully interact with prefix length, and the best-performing prefix lengths for DSQ7B remain effectively constant.

Table 12: PoLR is robust to different temperature samplings.

| $L_p$ | Sampling Temperatures | | | | |
|---|---|---|---|---|---|
| | 0.2 | 0.4 | 0.6 | 0.8 | 1 |
| 1 | 88.0 | 89.4 | 89.2 | 89.2 | 89.0 |
| 2 | 88.0 | 89.4 | 89.2 | 89.2 | 88.8 |
| 4 | 88.2 | 89.4 | 89.0 | 88.6 | 88.2 |
| 8 | 88.2 | 89.2 | 89.2 | 88.8 | 88.0 |
| 16 | 88.2 | 89.6 | 89.0 | 88.8 | 89.4 |
| 32 | 88.2 | 89.2 | 89.4 | 89.0 | 89.2 |
| 64 | 88.0 | 89.0 | 89.0 | 89.4 | 88.4 |
| 128 | 87.6 | 89.4 | 89.0 | 88.6 | 88.6 |
| 256 | 88.4 | 89.4 | 89.4 | 88.6 | 88.0 |
| 512 | 88.4 | 90.0 | 89.6 | 89.4 | 89.0 |
| 1024 | 88.2 | 88.6 | 89.2 | 89.0 | 88.6 |
| 2048 | 88.2 | 89.4 | 89.2 | 89.0 | 88.6 |
| 4096 | 88.6 | 89.6 | 89.2 | 88.6 | 88.4 |
| SC | 88.0 | 89.4 | 89.2 | 89.2 | 88.8 |

