# OpenReview forum: "THE PATH OF LEAST RESISTANCE: GUIDING LLM REASONING TRAJECTORIES WITH PREFIX CONSENSUS"
_ICLR.cc/2026/Conference — ICLR 2026 Poster_

### Official Review · Reviewer_jggQ · 2025-10-27

**Soundness:** 2
**Presentation:** 3
**Contribution:** 2
**Rating:** 6
**Confidence:** 2

**Summary:**

This paper proposes an inference method called PoLR to address the issue of high computational costs in the inference process of LLM. Traditional Self-Consistency decoding requires sampling and expanding multiple complete reasoning paths, which improves accuracy but incurs significant computational overhead. The core idea of PoLR is to utilize prefix consistency during inference: first, generate short prefixes of multiple reasoning paths, cluster them, and only expand the paths in the main clusters, thereby significantly reducing ineffective computations. This method does not require retraining the model and can be used as a plug-in during the inference stage. The main contributions of the paper include: proposing the first method that leverages prefix consistency to optimize self-consistency decoding in the inference stage, reducing the number of generated tokens by up to approximately 60% and inference latency by about 50% without compromising accuracy. Additionally, the authors explain through theoretical analysis from an information-theoretic perspective why early reasoning prefixes contain strong signals for predicting the final correct answer, and verify the effectiveness of the PoLR method on mathematical, common sense, and scientific reasoning benchmarks through extensive experiments.

**Strengths:**

This paper introduces a highly novel inference strategy. Unlike existing work that either requires model fine-tuning or adaptively halts sampling based on answer agreement, PoLR introduces clustering based on early-stage reasoning prefixes. This is the first method to leverage prefix consistency at inference time for self-consistency decoding. The proposed idea is distinct from prior approaches and presents a fresh, efficient angle for improving LLM inference.

The theoretical analysis is solid and well-motivated. Using mutual information and conditional entropy, the paper shows that prefix clusters carry predictive signals about final answer correctness. It introduces the notion of cluster skew and formally proves that higher skew leads to greater token efficiency. The paper successfully connects theory and practice, and the theoretical results help explain the method’s robustness and efficiency.

The experiments are comprehensive and convincing. PoLR is evaluated on GSM8K, MATH500, AIME24/25, and GPQA-DIAMOND, covering a broad spectrum of reasoning difficulty. It is tested on LLMs ranging from 1.5B to 32B parameters and across different architectures and training paradigms. Results consistently show 40–60% token savings without accuracy degradation—sometimes even improving accuracy. PoLR is also shown to complement adaptive methods, offering additional savings. Results are averaged over 10 runs with reported standard deviations, ensuring statistical reliability. This section is a strong point of the paper.

**Weaknesses:**

Limitations of the method's applicability: There is a high degree of consensus in the early steps of PoLR hypothesis reasoning, meaning that the prefixes of most sampling paths converge to the main pattern. However, this assumption may not be fully valid for certain tasks. If the problem-solving paths of a question are highly divergent or the reasoning path corresponding to the correct answer is not the mainstream pattern, PoLR may miss important reasoning branches, thereby affecting accuracy.

Dependency on hyperparameters and clustering strategies: PoLR needs to pre-set the prefix length Lp and the methods for embedding and clustering prefixes. The optimal prefix truncation length and clustering method may vary across different tasks. If the prefix is too short, it may fail to capture sufficient reasoning signals; if it is too long, it may introduce irrelevant information and affect the clustering effect. In addition, simple word frequency embedding may not cover diverse expressions with semantic equivalence. Although the authors tried dense vector embedding and pointed out that its gain is limited and the overhead is large, the parameter selection in the clustering step itself still needs to be carefully adjusted in practical applications to ensure robustness.

**Questions:**

Generality to Non-Math Tasks: Most evaluations are on math, STEM, and QA-style reasoning. Have you tested PoLR on more open-ended tasks where prefix similarity may be weaker?

Voting within Noisy Clusters: When the dominant cluster includes low-quality traces, SC’s  majority voting might fail. Has the team considered weighting cluster votes by internal consistency or agreement confidence?

Failure Cases: While PoLR generally preserves accuracy, there are cases with slight degradation. Can the authors provide qualitative analysis or examples to understand what types of reasoning problems lead to this drop?

Cluster Selection Robustness: PoLR selects the dominant cluster for expansion. In settings where reasoning traces diverge meaningfully, how sensitive is performance to this choice? Did you try expanding the top-k clusters instead of only the largest one?

---

> ### Author Response · Authors · 2025-11-22
> **Response to Weakness 1 and Question 1**
>
> We would to thank the reviewer for the detailed assessment of our work, particularly regarding the novelty of PoLR, and the breadth and rigor of the experiments. We sincerely appreciate your thoughtful reading.
>
> >**[W1, Q1] Limitations of the method's applicability on Math/STEM datasets**
>
> Correct indeed. In section 2 of the paper we conducted an analysis over different Math and reasoning datasets and observed that the reasoning traces with the correct answers tend to exhibit high similarity in the early stages of the generation process. However, we agree with the reviewer that the datasets which are more open-ended or involve creative reasoning may not follow this trend. If the reviewer have a certain task and dataset in mind please let us know, we will perform experiments on that task and will include in the paper.
>
> Additionally, we have experimented with a non-math/STEM QA task, **StrategyQA**. We observe the PoLR gains hold true for this dataset as well. Here is the PoLR comparison with SC on **StrategyQA** using DSQ-7B ($N=51$).
>
> | Prefix length | SC(%) | PoLR(%) | $PExp$(%) | $\eta$(%) | $K_t$(ms) |
> | -| -| -| -| -| -|
> |256|59.8|59.6|66.2|31.5|9.4|
> |512|59.8|59.8|65.7|15.0|12.2|
> |1024|59.8|61.1|65.5|6.3|12.1|
> |2048|59.8|60.5|65.1|5.8|13.0|
> |4096|59.8|61.8|65.6|6.1|13.1|
>
> We have included an extended version of this table with more prefix lengths in the additional dataset experiments *Appendix F Table 10*.

---

> ### Author Response · Authors · 2025-11-22
> **Response to Weakness 2**
>
> >**[W2, Q2] Dependency on hyperparameters and clustering strategies:**
>
> We appreciate the reviewer for raising these concerns. We shall aim to provide an explanation on 3 grounds:
> - Choice of Prefix Length
> - Choice of Clustering Method
> - Choice of Cluster selection strategy for expansion
>
> ### Choice of Prefix Length
> We agree with your concerns w.r.t. the optimal choice of prefix length and would like to highlight that in our observations, the choice of prefix length is more of a trade-off between the number of traces expanded and the token efficiency provided by the method. Further, as revealed in *Figure 4*, we notice that across a wide range of tasks and model combinations, prefix length $L_p = 256$ proves to be a reasonable default setting providing a balance between token efficiency and accuracy.
> For reference, we also attach below, the accuracies across prefix lengths for various dataset-model combinations:
>
> | |GSM8K_DSQ15B|Math500_DSQ7B|AIME24_PHI15B|AIME25_DSQ7B|GPQA_QWQ32B|
> |-|-|-|-|-|-|
> |SC|73.16|89.80|50.00|33.33|68.69|
> |64|72.63|89.80|56.67|33.33|68.69|
> |128|72.25|89.20|50.00|36.67|68.69|
> |**256**|**73.16**|**89.40**|**53.33**|**33.33**|**70.20**|
> |512|72.93|90.00|50.00|36.67|67.68|
> |1024|72.86|89.20|50.00|36.67|67.68|
> |2048|73.16|90.40|50.00|33.33|69.19|
>
> - W.r.t shorter prefix lengths, we expected it to fail. Empirically, we found that even shorter prefix lengths (e.g., 2) provide accuracies comparable to SC, but at the cost of token efficiency.
> - W.r.t longer prefix lengths, we expected it to perform at par with SC, but we found that the improvement in accuracy is marginal, while the token efficiency is significantly reduced.
>
> Therefore, 256 strikes a reasonable balance between the two extremes, and hence we chose it as the default setting.
>
> ### Choice of Clustering Method
> We have added a detailed section discussing the choice of clustering methods and their configurations, including further experiments performed with **new clustering techniques in Section 5.4 Table 5** and we have included PoLR robustness to other hypeparameters in *Appendix I*.
>
> For reference, the following tables showcase the accuracy and efficiency of different clustering methods for Math500 dataset.
>
> |  | SC | Agglomerative |   | DBSCAN  |   | HDBSCAN |   |
> | -| -| -| -| -| -| -| -|
> | Model   | Acc | Acc | $\eta$ | Acc | $\eta$ | Acc | $\eta$ |
> | DSQ1.5B | 76.2 | 75.4 | 52.4| 76.2| 68.7 | 76.0 | 67.4|
> | DSQ7B   | 89.8| 89.4 | 48.7| 89.6| 54.6| 89.2| 63.5 |
> | QWQ32B  | 91.8| 92.0| 51.8| 92.0| 71.3 | 92. | 63.1
>
> Results on other datasets GSM8k and GPQA are added in *Section 5.4, Table 5*. The results demonstrate that the **gains from PoLR persist, irrespective of the clustering technique**, with minor variations in accuracy.
>
> ### Choice of Cluster selection strategy for expansion
>
> We have added a detailed section discussing the choice of cluster selection strategy for expansion, in *Section 5.6, Table 6*. Specifically, here we present results from experiments conducted using DSQ-7B ($L_p=256$,$N=51$)(with clusters arranged in decreasing order of dominance):
>
> | Cluster| Math500| GSM8K| GPQA|
> | - | -| - | - |
> | 0 | 89.4| 80.2| 55.6|
> | 1 | 89.2| 79.4| 51.5|
> | 2 | 78.4| 64.5| 47.5|
> | 3 | 45.6| 34.3| 34.3|
> ...
>
> As we can see from these results, the **accuracy of the dominant cluster is consistently higher** than the other clusters.  Further we compute the Pearson correlation coefficient between cluster size and accuracy for DSQ7B and QwQ32B ($N=51$,$L_p=256$) for all the datasets.
>
> | Model | GSM8K | MATH500 | AIME24 | AIME25 | GPQA |
> | - | - | - | - | - | - |
> | DSQ7B  | 0.90| 0.89| 0.75| 0.76| 0.90 |
> | QwQ32B | 0.90| 0.88| 0.87| 0.78| 0.78 |
>
> This clearly depicts the high correlation between the cluster size and accuracy ($\rho \geq 0.75$), suggesting that the dominant cluster indeed **captures most accurate traces**.
>
> We have incorporated all these tables in the paper. Thank you again for this insightful comments this analysis further support towards our claim that the cardinality based strategy for cluster selection works well in practice.

---

> ### Author Response · Authors · 2025-11-22
> **Responses to Questions**
>
> Responses to Question 1 and 2 are provided in the responses to the weaknesses.
>
> ---
> ---
>
> >**[Q3] Failure Cases:** While PoLR generally preserves accuracy, there are cases with slight degradation. Can the authors provide qualitative analysis or examples to understand what types of reasoning problems lead to this drop?*
>
> We acknowledge the 10% drop for QwQ32B on AIME2025. However, we would also like to highlight that as the dataset contains just 30 samples, 10% delta translates to a deviation in just 3 samples. Furthermore, analyzing these failure cases, we obtain the following interesting observations:
>
> |Case|Cluster Rank|Cluster Size|%Correct Traces Within Cluster|(SC Consensus %)|
> |- |- |- |- |- |
> |Failure Case 1| 0  | 20 | 45%  | 53% |
> | |1| 19|53%|  |
> | |2| 12|67%|  |
> | Failure Case 2 | 0  | 22 | 41%  | 59% |
> | |1| 15|73%|  |
> | |2| 14|71%|  |
> | Failure Case 3 | 0  | 20 | 45%  | 63% |
> | |1| 13|62%|  |
> | |2| 13|85%|  |
> | Average Case | 0  | 21.76| 66.05% | **67.91%** |
> | | 1| 14.63 | 68.54%|  |
> | | 2| 9.7| 65.26% |  |
>
> - For failure cases,  SC consensus ($correct traces/51) itself is much lower than the average case indicating that the failure cases are more challenging even for SC baselines.
> - When SC itself succeeds only marginally, PoLR is not expected to outperform it. We note this limitation in lines 303–305 and include the supporting results in *Appendix H, Table 11*.
>
> Despite PoLR failure on SC challenging problems, we want to highlight the PoLR's primary contribution is improving computational efficiency while maintaining SC-level accuracy on most instances.
>
>
> ---
> ---
>
> >**[Q4] Cluster Selection Robustness:** PoLR selects the dominant cluster for expansion. In settings where reasoning traces diverge meaningfully, how sensitive is performance to this choice? Did you try expanding the top-k clusters instead of only the largest one?*
>
> Thank you for raising this thought provoking question. To answer this question, we experimented with expanding multiple clusters for Math500 dataset, i.e. expanding all traces from more than one cluster and taking majority consensus from this *merged* cluster.
>
> | | DSQ-1.5B | | DSQ-7B |  | QwQ-32B |  |
> | -| -| -|- | -| -| -|
> | Clusters Expanded|Acc| Paths Expanded| Acc | Paths Expanded | Acc| Paths Expanded |
> | Top 1 | 75.8% | 22.3  | 89.4%  | 22.6  | 92.0%| 22.8  |
> | Top 2 | 76.0% | 38.0  | 89.4%  | 38.4  | 92.0%| 38.7  |
> | Top 3| 76.4% | 46.7  | 89.6%  | 46.7  | 91.8%| 47.1  |
> | Top 4  | 76.4% | 50.1  | 89.8%  | 50.0  | 91.8%| 50.3  |
> | Top 5 | 76.2% | 50.9  | 89.8%  | 50.9  | 91.8%| 50.9  |
> | Top 6 | 76.2% | 51.0  | 89.8%  | 51.0  | 91.8%| 51.0  |
> | Top 7 | NA | NA | 89.8%  | 51.0  | NA| NA |
> | SC| 76.2% | 51 | 89.8%  | 51 | 91.8%| 51 |
>
> Based on the above results, we observe the following:
> 1. **Expanding multiple clusters yields minimal accuracy gains** over the dominant cluster alone.
> 2. **Path expansions increase dramatically with additional clusters**, eliminating most efficiency benefits.
> 3. Finally, **expanding all clusters, converges to baseline SC**.

---

### Official Review · Reviewer_jah9 · 2025-10-31

**Soundness:** 3
**Presentation:** 3
**Contribution:** 2
**Rating:** 6
**Confidence:** 2

**Summary:**

The paper introduces PoLR (Path of Least Resistance), an inference-time method that improves the efficiency of reasoning in large language models (LLMs) without sacrificing accuracy. The key idea is to exploit prefix consistency, i.e., the observation that early reasoning steps (prefixes) across different reasoning traces tend to converge before diverging.

**Strengths:**

1. The paper builds on a compelling empirical finding: early reasoning steps in LLMs already encode signals predictive of correctness. This observation is both intuitively appealing and empirically validated, where clustering short prefixes achieves nearly identical accuracy to full SC.
2. The paper notes some accuracy drops (especially on AIME24/25) but doesn’t deeply analyze why PoLR fails there. Qualitative error analyses or visualizations of cluster distributions could clarify how prefix diversity correlates with performance loss.

**Weaknesses:**

1. Uses simple TF–IDF clustering, which may miss deeper semantic similarities between reasoning traces.

2. Accuracy drops on some datasets (e.g., AIME24/25) are not fully explored or explained.

3. Mutual information argument is mostly qualitative and lacks empirical validation.

4. Clustering approach may not scale efficiently when many samples (large N) are generated.

5. Some sections are dense, with mixed implementation and theoretical details; figures could be clearer.

**Questions:**

N/A

---

> ### Author Response · Authors · 2025-11-22
>
> First of all, We thank you for your thoughtful and constructive review. We appreciate you recognizing core empirical contribution and the positive assessment of the overall approach.
>
> > **[W1] Deeper Semantics** Uses simple TF–IDF clustering, which may miss deeper semantic similarities between reasoning traces.
>
> We acknowledge the reviewer's concern regarding the TF-IDF encoding strategy. We would like to highlight the results of the comparison between the TF-IDF and Semantic Embedding strategies are compared in *Table 4 Section 5.2*. We apologize for any unintended ambiguity in our presentation.
>
> Further, we have also expanded the evaluation comparison to more datasets, including MATH500 and GPQA for different language models and updated the *Table 4* accordingly. Following are the results for MATH500:
>
> | | | TF-IDF | | | Semantic |(Matryoshka) | |
> | -| -| -| -|- | -| -|- |
> | | SC |  ACC | $\eta$  | $K_t$ | ACC| $\eta$ | $K_t$ |
> | DSQ1.5B | 76.2 | 75.4 | 52.4| 6.5 | 76.2 | 47.5| 219.7 |
> | DSQ7B | 89.8 | 89.4 | 48.7| 7.6 | 90.0 | 44.7| 220.6 |
> | QWQ32B | 91.8 | 92.0 | 51.8| 11.2  | 91.8 | 51.4| 218.9 |
>
> Once again, we observe that **semantic embedding techniques offer minimal to no accuracy gains despite higher overall computational overhead**. Hence, our recommended default strategy remains as TF-IDF for PoLR.
>
> ---
> ---
>
> >**[W2] Instance-level error analysis** Accuracy drops on some datasets (e.g., AIME24/25) are not fully explored or explained.
>
> We acknowledge the 10% drop for QwQ32B on AIME2025. However, we would also like to highlight that as the dataset contains just 30 samples, 10% delta translates to a deviation in just 3 samples. Furthermore, analyzing these failure cases, we obtain the following interesting observations:
>
> |Case|Cluster Rank|Cluster Size|%Correct Traces Within Cluster|(SC Consensus %)|
> |- |- |- |- |- |
> |Failure Case 1| 0  | 20 | 45%  | 53% |
> | |1| 19|53%|  |
> | |2| 12|67%|  |
> | Failure Case 2 | 0  | 22 | 41%  | 59% |
> | |1| 15|73%|  |
> | |2| 14|71%|  |
> | Failure Case 3 | 0  | 20 | 45%  | 63% |
> | |1| 13|62%|  |
> | |2| 13|85%|  |
> | Average Case | 0  | 21.76| 66.05% | **67.91%** |
> | | 1| 14.63 | 68.54%|  |
> | | 2| 9.7| 65.26% |  |
>
> - For failure cases,  SC consensus ($correct traces/51) itself is much lower than the average case indicating that the failure cases are more challenging even for SC baselines.
> - When SC itself succeeds only marginally, PoLR is not expected to outperform it. We note this limitation in lines 303–305 and include the supporting results in *Appendix H, Table 11*.
>
> Despite PoLR failure on SC challenging problems, we want to highlight the PoLR's primary contribution is improving computational efficiency while maintaining SC-level accuracy on most instances.
>
> ---
> ---
>
> >**[W3]** Mutual information argument is mostly qualitative and lacks empirical validation.
>
> Thank you for this observation. Due to page length restrictions we could not link the theoretical justifications with the empirical evidence. Corresponding plots for GSM8K (evaluated using DSQ1.5B, DSQ7B, Qwen-Math-7B) are present in Section 5.5 Figure 3. Further we have added the raw numbers in tabular format in Appendix E Table 9, one of the cases is presented below for GSM8K DSQ7B for different prefix lengths.
>
> |$L_p$|path_efficiency| avg_skew|avg_nmi|
> | - | - | - | - |
> | 256 |0.52| 0.46|0.142|
> | 512|0.53| 0.47| 0.150 |
> | 1024|0.53 | 0.47| 0.151 |
> | 2048|0.53| 0.47| 0.150|
> | 4096|0.53| 0.47| 0.151|
>
> ---
> ---
>
> >**[W4]** Clustering approach may not scale efficiently when many samples (large N) are generated.
>
> We acknowledge the concern regarding the super-linear asymptotic time-complexity of the clustering algorithm. However, we would like to offer an alternative perspective on this concern. PoLR employs the prefix encoding and clustering step ONCE during the entire generation process. However, the majority compute overhead is still tied to the autoregressive generation process of the language model. For example, in our experiments, we expand 51 traces in line with existing research works employing Self-Consistency adaptations, resulting in an inference time overhead in the order of ~10ms.
>
> As a result, for the clustering overhead to have a noticeable drawback, the number of samples (i.e., the LLM's output traces) needed to be generated would be far greater than what is typically required for the task at hand, for example: following are the max number of traces expanded by existing leading research works:
> - SC (Wang et al. (2023)): 40
> - AC (Aggarwal et al. (2023)) : 40
> - ESC (Li et al. (2024)): 64 (MATH500), 40 otherwise
>
> Given that we expand only 51 traces—fully in line with prior SC-based work—the clustering step remains negligible compared to the cost of autoregressive generation
>
> ---
> ---
>
> > **[W5] Presentation and clarity**
>
> We appreciate the feedback. In response: we have mode “prefix”  definitions more explicitly. Regarding Figures: we are working on it and will provide these plots as supplementary material.

---

### Official Review · Reviewer_yLoa · 2025-10-31

**Soundness:** 3
**Presentation:** 1
**Contribution:** 2
**Rating:** 6
**Confidence:** 2

**Summary:**

The authors propose PoLR, a method to reduce the computational cost of SC decoding in LLMs. Instead of expanding all sampled reasoning paths, PoLR clusters short prefixes and expands only those in the dominant cluster. The method is shown to preserve accuracy while drastically reducing token usage and latency.

**Strengths:**

1. The paper conducts comprehensive experiments to demonstrate the effectiveness of the proposed method. The main results show that it consistently outperforms existing approaches.

2. The idea is both novel and compelling, and PoLR is straightforward to implement and fully compatible with existing language models.

**Weaknesses:**

I don't have many comments regarding the weaknesses; however, the primary issue is that the paper is not well written and is difficult to follow. For instance, I had to consult referenced papers to understand prerequisite concepts such as the definition of a prefix and the detailed observations related to prefix consistency. It would be better if the paper were more self-contained. Nonetheless, given the strong experimental results, I lean toward a positive assessment of the paper.

**Questions:**

NA

---

> ### Author Response · Authors · 2025-11-22
>
> Thank you very much for the thoughtful review and for highlighting both the novelty of PoLR and the strength of our experimental results.
>
> > **Presentation quality**
>
> We appreciate your feedback on the writing. We have added the relevant definitions such as "prefix" at line number 044 in the modified version and proper connection with the referenced papers. We will keep making changes to improve the presentations of the paper.
>
> > **More experiments to further strengthen PoLR**
>
> We have added additional tables and plots to better illustrate the better illustrate the behavior and robustness of PoLR. These include:
> - **Clustering method comparison-** We evaluate alternative clustering techniques such as DBSCAN and HDBSCAN.
>     - The new results on GSM8K, MATH500, and GPQA (*Section 5.4, Table 5*) show that PoLR’s gains persist across clustering methods, with only minor variations in accuracy.
> - **Extended semantic-embedding experiments-** We broadened the embedding-based analysis to additional datasets, including MATH500 and GPQA
>     - On these datasets as well, we observe that semantic embedding offer minimal to no accuracy gains despite higher overall computational overhead (*Section 5.2, Table 2*). Thus strengthening our recommended default strategy as TF-IDF for PoLR.
> - **Sensitivity analysis to LLM temperature-** We conducted more analysis on the impact of different temperature sampling wrt prefix length.
>     - We observe that **PoLR remains stable across these settings** and continues to provide consistent improvements in *Appendix I.*
> - **New dataset (Non-STEM)-** To evaluate generality beyond mathematical reasoning, we added experiments on StrategyQA, a commonsense reasoning dataset.
>     - PoLR again shows strong and robust performance in this domain in *Appendix F, table 10*
>
> These additions provide a more complete and self-contained view of PoLR’s behavior and further support its effectiveness across diverse tasks and configurations.

---

> ### Author Response · Authors · 2025-11-25
> **Presentation improvements**
>
> Dear reviewer, we further improved the writing and made the paper more self-contained by adding clear definitions of prefixes, prefix consistency, and related ideas directly in the introduction. We have also simplified the exposition and clarified key transitions.
>
> Thank you for your presentation suggestion and help us make the paper much easier to read without relying on external references.
>
>
> To be precise, we have updated the Section 1 Introduction with these two paragraphs
>
> ```
> To reduce SC’s compute requirements, several inference-time methods such as Adaptive Consistency (AC) Aggarwal et al. (2023) and Early-Stop Self-Consistency (ESC) Li et al. (2024) have been proposed. These methods expand reasoning traces sequentially and stop generating them only when sufficient final-answer agreement is observed. Though effective, they share a fundamental limitation: answer-level agreement is only observable after full reasoning traces is generated. As a result, they cannot exploit the rich structural information that might appear earlier in the reasoning process and their efficiency remains limited by the need to generate complete reasoning traces.
>
> Recently, an alternative line of research shows that the early stages of reasoning traces carry disproportionately strong signals about the eventual solution, a phenomenon known as prefix consistency. Formally, if $r_i$ denotes a reasoning trace, then its first L tokens $r_i[1 : L]$, termed as prefix, tend to exhibit similarity across reasoning traces, irrespective of their later steps. Ji et al. (2025) exploited this phenomenon at training time, that is, fine-tuning models on prefixes to improve reasoning while reducing inference cost. However, this requires expensive fine-tuning and cannot be applied directly at inference.
> ```

---

### Official Review · Reviewer_w1dY · 2025-11-01

**Soundness:** 3
**Presentation:** 3
**Contribution:** 3
**Rating:** 6
**Confidence:** 3

**Summary:**

This paper introduces PoLR, an inference-time optimization that tackles the high computational cost of Self-Consistency (SC). PoLR generates short reasoning prefixes, clusters them to find a consensus, and only extends the paths from the dominant cluster. This approach significantly reduces token usage and latency on complex reasoning tasks while maintaining or even improving accuracy.

**Strengths:**

- Innovative & Practical: Cleverly solves the high cost of SC by using prefix consensus to filter paths early, a novel and practical approach.
- High Efficiency: Achieves impressive results, reducing token/latency costs by ~50% without sacrificing accuracy.
- Well-Supported: Backed by solid theoretical analysis (information theory, structural skew) and extensive experiments across multiple models and benchmarks.
- Plug-and-Play: A simple, training-free method that is easy to implement and complements other optimization techniques.

**Weaknesses:**

- While the MI and skew analyses are insightful, they are not empirically validated. Quantitative measures linking $I(Z;Y)$ to observed behavior are missing.
- Cluster-Dependent: The method's success hinges on identifying the correct dominant cluster; it could potentially filter out correct but less common reasoning paths.
- Limited on Certain Tasks: Shows weaker performance on tasks with low lexical overlap (e.g., GPQA-DIAMOND), where the prefix consistency signal is faint.
- Hyperparameter Discussion: Could benefit from a deeper analysis of the clustering method choice and TF-IDF configurations.

**Questions:**

1. If the dominant cluster leads to incorrect reasoning, does PoLR amplify “high-confidence errors”? Could multi-cluster expansion or confidence estimation help?
2. Could the prefix length be adjusted dynamically via sampling temperature for adaptive balance between easy and hard problems?

---

> ### Author Response · Authors · 2025-11-22
> **Responses to observed weaknesses**
>
> We thank the reviewer for providing thoughtful feedback. Below, we provide detailed responses.
>
> > **[W1] NMI empirical validation**: While the MI and skew analyses are insightful, they are not empirically validated. Quantitative measures linking to observed behavior are missing.
>
> Thank you for this observation. Due to page length restrictions we could not link the theoretical justifications with the empirical evidence. Corresponding plots for GSM8K (evaluated using DSQ1.5B, DSQ7B, Qwen-Math-7B) are present in Section 5.5 Figure 3. Further we have added the raw numbers in tabular format in Appendix E Table 9, one of the cases is presented below for GSM8K DSQ7B for different prefix lengths.
>
> | $L_p$  | path_efficiency | avg_skew | avg_nmi |
> | - | - | - | -  |
> | 256 | 0.52| 0.46| 0.142|
> | 512| 0.53| 0.47| 0.150 |
> | 1024| 0.53 | 0.47| 0.151 |
> | 2048| 0.53| 0.47| 0.150|
> | 4096| 0.53| 0.47| 0.151|
>
> ---
> > **[W2] Cluster-Dependence** The method's success hinges on identifying the correct dominant cluster; it could potentially filter out correct but less common reasoning paths.
>
> We agree that in certain cases, it is possible that the dominant cluster does not fully capture the most accurate traces. In order to check the validity of this hypothesis, **we analyzed the cluster-wise accuracies** across a range of datasets, and have added this analysis in Section 5.6 Table 6.
>
> Specifically, here we present results from experiments conducted using DSQ-7B ($L_p=256$,$N=51$)(with clusters arranged in decreasing order of dominance):
>
> | Cluster| Math500| GSM8K| GPQA|
> | - | -| - | - |
> | 0 | 89.4| 80.2| 55.6|
> | 1 | 89.2| 79.4| 51.5|
> | 2 | 78.4| 64.5| 47.5|
> | 3 | 45.6| 34.3| 34.3|
> ...
>
> As we can see from these results, the **accuracy of the dominant cluster is consistently higher** than the other clusters.  Further we compute the Pearson correlation coefficient between cluster size and accuracy for DSQ7B and QwQ32B ($N=51$,$L_p=256$) for all the datasets.
>
> | Model | GSM8K | MATH500 | AIME24 | AIME25 | GPQA |
> | - | - | - | - | - | - |
> | DSQ7B  | 0.90| 0.89| 0.75| 0.76| 0.90 |
> | QwQ32B | 0.90| 0.88| 0.87| 0.78| 0.78 |
>
> This clearly depicts the high correlation between the cluster size and accuracy ($\rho \geq 0.75$), suggesting that the dominant cluster indeed **captures most accurate traces**.
>
> We have incorporated all these tables in the paper. Thank you again for this insightful comments this analysis further support towards our claim that the cardinality based strategy for cluster selection works well in practice.
>
> ---
> > **[W3] Weaker performance on GPQA** Shows weaker performance on tasks with low lexical overlap (e.g., GPQA-DIAMOND), where the prefix consistency signal is faint.
>
> Thank you for the observation. To examine PoLR’s lower performance on GPQA-Diamond, we analyzed the clustering behavior of prefixes across datasets. Despite the task’s low lexical overlap, we find that the prefix consistency signal remains stable: the cluster size distributions for GPQA-Diamond closely match those for GSM8K and Math500.
>
> For DSQ7B at prefix length 256, the cluster sizes across tasks are:
>
> | Cluster| Math500 | GSM8K | GPQA  |
> | -| -| -| -|
> |0 | 22.6 | 23.95 | 20.79 |
> | 1 | 15.8 | 16.03 | 15.81 |
> | 2 | 8.27 | 7.74  | 9.47  |
> |  3 | 3.29| 2.73  | 4.03  |
> ...
>
> These values indicate that the clusters formed for GPQA are comparable in size to those formed for other tasks, suggesting that the **prefix consistency signal for GPQA is still strong.** The reduced performance likely stems from the inherent difficulty of the dataset rather than weaker prefix consistency.
>
> ---
> > **[W4] Hyperparameter Discussion** Could benefit from a deeper analysis of the clustering method choice and TF-IDF configurations.
>
> We appreciate your suggestion regarding hyperparameter discussion. We have added a detailed section discussing the choice of clustering methods and their configurations, including further experiments performed with **new clustering techniques in Section 5.4 Table 5** and we have included PoLR robustness to other hypeparameters in *Appendix I*.
>
> For reference, the following tables showcase the accuracy and efficiency of different clustering methods for Math500 dataset.
>
> |  | SC | Agglomerative |   | DBSCAN  |   | HDBSCAN |   |
> | -| -| -| -| -| -| -| -|
> | Model   | Acc | Acc | $\eta$ | Acc | $\eta$ | Acc | $\eta$ |
> | DSQ1.5B | 76.2 | 75.4 | 52.4| 76.2| 68.7 | 76.0 | 67.4|
> | DSQ7B   | 89.8| 89.4 | 48.7| 89.6| 54.6| 89.2| 63.5 |
> | QWQ32B  | 91.8| 92.0| 51.8| 92.0| 71.3 | 92. | 63.1
>
> Results on other datasets GSM8k and GPQA are added in Section 5.4 Table 5. The results demonstrate that the **gains from PoLR persist, irrespective of the clustering technique**, with minor variations in accuracy.
>
> Following the reviewer's insightful suggestion, we observed that the newly tested clustering techniques can provide better token efficiency, however, the choice of hyperparameters for these techniques is to be studied.

---

> ### Author Response · Authors · 2025-11-22
> **Responses to the Questions**
>
> >**[Q1] Multi-cluster Expansion** If the dominant cluster leads to incorrect reasoning, does PoLR amplify “high-confidence errors”? Could multi-cluster expansion or confidence estimation help?
>
>
> Thank you for raising this thought provoking question. To answer this question, we experimented with expanding multiple clusters for Math500 dataset, i.e. expanding all traces from more than one cluster and taking majority consensus from this *merged* cluster.
>
> |                       | DSQ-1.5B |                   | DSQ-7B |                   | QwQ-32B |                   |
> | --------------------- | -------- | ----------------- | ------ | ----------------- | ------- | ----------------- |
> | Clusters Expanded | Acc      | Paths Expanded | Acc    | Paths Expanded | Acc     | Paths Expanded |
> | Top 1 | 75.8%    | 22.3              | 89.4%  | 22.6              | 92.0%   | 22.8              |
> | Top 2 | 76.0%    | 38.0              | 89.4%  | 38.4              | 92.0%   | 38.7              |
> | Top 3| 76.4%    | 46.7              | 89.6%  | 46.7              | 91.8%   | 47.1              |
> | Top 4  | 76.4%    | 50.1              | 89.8%  | 50.0              | 91.8%   | 50.3              |
> | Top 5                     | 76.2%    | 50.9              | 89.8%  | 50.9              | 91.8%   | 50.9              |
> | Top 6                     | 76.2%    | 51.0              | 89.8%  | 51.0              | 91.8%   | 51.0              |
> | Top 7                     | NA       | NA                | 89.8%  | 51.0              | NA      | NA                |
> | SC                    | 76.2%    | 51                | 89.8%  | 51                | 91.8%   | 51                |
>
> Based on the above results, we observe the following:
> 1. **Expanding multiple clusters yields minimal accuracy gains** over the dominant cluster alone.
> 2. **Path expansions increase dramatically with additional clusters**, eliminating most efficiency benefits.
> 3. Finally, **expanding all clusters, converges to baseline SC**.
>
> Further, regarding *confidence* criteria, in our current implementation, the cluster expansion strategy does not employ any confidence estimation criterion and confidence-guided top-K cluster expansion poses a promising next direction to explore.
> We note that when expanding multiple clusters, the accuracy gains are minimal but the token efficiency gains are substantially reduced, hence Impacting the major motivation of PoLR.
>
> ---
> ---
>
> >**[Q2] Prefix-length Vs Temperature sampling** Could the prefix length be adjusted dynamically via sampling temperature for adaptive balance between easy and hard problems?
>
> Thank you for your insightful suggestion. We have experimented PoLR over a combination of prefix length Vs. the sampling temperature on MATH500 with DSQ7B LLM ($N=51$) and observed the following:
>
> |      | 0.2   | 0.4   | 0.6   | 0.8   | 1     |
> | ---- | ----- | ----- | ----- | ----- | ----- |
> | 32   | 88.2% | 89.2% | 89.4% | 89.0% | 89.2% |
> | 64   | 88.0% | 89.0% | 89.0% | 89.4% | 88.4% |
> | 128  | 87.6% | 89.4% | 89.0% | 88.6% | 88.6% |
> | 256  | 88.4% | 89.4% | 89.4% | 88.6% | 88.0% |
> | 512  | 88.4% | 90.0% | 89.6% | 89.4% | 89.0% |
> | 1024 | 88.2% | 88.6% | 89.2% | 89.0% | 88.6% |
> | 2048 | 88.2% | 89.4% | 89.2% | 89.0% | 88.6% |
> | 4096 | 88.6% | 89.6% | 89.2% | 88.6% | 88.4% |
> | SC   | 88.0% | 89.4% | 89.2% | 89.2% | 88.8% |
>
> Based on the above experiment, we do not notice major improvements in accuracy when adjusting the prefix length based on the sampling temperature. This suggests that the **optimal prefix length do not vary significantly across different sampling temperatures**, and that a fixed prefix length may be sufficient for achieving good performance across a range of sampling temperatures.
>
> Therefore, dynamic prefix length adjustment may not justify the additional complexity and computational overhead, and a fixed prefix length may be a simpler and more effective approach for balancing between easy and hard problems. We have added this analysis to the paper in *Appendix I*.

---

### Author Response · Authors · 2025-11-26
**Global Revision Summary**

Dear reviewers,

Thank you for your constructive feedback and thorough review. We are excited to share that your valuable feedback helped us to significantly improve our manuscript. Here is summary of the  revisions we have made.

### Additional Experiments
We have conducted several additional experiments and have added corresponding results into the paper to better illustrate the behavior and robustness of PoLR. These include:
- **Clustering method comparison** ==========> [Reviewer: w1dY]
    - *Experiments*: In addition to current agglomerative clustering, we have evaluated PoLR with alternative clustering techniques such as DBSCAN and HDBSCAN to test PoLR robustness to different clustering methods.
    - *Results*: The new results on GSM8K, MATH500, and GPQA  show that `PoLR’s gains persist across clustering methods`, with only minor variations in accuracy.
    - *Action*:  We have added this analysis in main paper **Section 5.4, Table 5**.
- **Extended semantic-embedding experiments** ==========> [Reviewer: w1dY,  jah9]
    - *Experiments*: We broadened the embedding-based analysis to additional datasets, including MATH500 and GPQA  to understand TF-IDF generalizability to different datasets in
    - *Results*: We observe that `semantic embedding offer minimal to no accuracy gains despite higher overall computational overhead`. Thus strengthening our recommended default strategy as TF-IDF for PoLR.
    - *Action*: We have enhanced **Section 5.2, Table 2** with Matroyshka embeddings evaluation on GSM8K, Math500 and GPQA-Diamond datasets.
- **Sensitivity analysis to LLM temperature**  ==========> [Reviewer: jggQ]
    - *Experiments*: We conducted additional analysis on the impact of different temperature sampling w.r.t varied prefix length.
    - *Results*: We observe that `PoLR remains stable across different temperature settings`.
    - *Action*: We have added this analysis in **Appendix I, Table 12.**
- **New datasets (Non-STEM)**  ==========> [Reviewer: w1dY,  jggQ]
    - *Experiments*: To evaluate generality beyond mathematical reasoning, we conducted PoLR experiments on StrategyQA, a multi-hop commonsense reasoning dataset.
    - *Results*: PoLR consistently shows `strong and robust performance on non-STEM tasks`.
    - *Action*: We briefly mention about this in the main paper in Section 4 Main results. We have also added the full experiment table in **Appendix F, Table 10**
- **Instance-level Error analysis** ==========> [Reviewer:  jah9,  jggQ]
    - *Experiments*: To understand why PoLR fails on some cases where SC passes, we conducted an instance level analysis on AIME25 datasets.
    - *Results*: We find that for the challenging problems where even the SC baseline solves the task marginally, PoLR is not expected to outperform SC in accuracy. However, `the main gain of the PoLR in terms of token efficiency remains unaffected`.
    - *Action*: We have talked about this in Section 4 main results and added this complete analysis in **Appendix H, Table 11.**


### Paper Presentation Improvement

[Reviewer: yLoa,  jah9]

We have improved the writing and made the paper more self-contained by adding clear definitions of "prefixes", "prefix consistency", and related ideas directly in the introduction. We have also simplified the exposition and clarified key transitions. We also have improved narrative flow in Section 2 by transforming it from an experiment-heavy description to a preliminary study framework.


Overall, these additions provide a more complete and self-contained view of PoLR’s behavior and further support its effectiveness across diverse tasks and configurations. We appreciate insightful comments from all reviewers.Detailed responses to specific reviewer questions follow in subsequent comments.

---

### Meta-Review · Area_Chair_LDMJ · 2026-01-07

**Summary:**

Reviewers generally view PoLR as a simple, practical inference-time variant of self-consistency (cluster short reasoning prefixes, expand only the dominant cluster) with significant token savings and largely maintained accuracy. All reviews give a score of 6: the main hesitation is whether the method will prune minority-but-correct solution paths and whether the paper’s results can generalize beyond STEM-style reasoning. Additional concerns include sensitivity to design choices (prefix length, clustering method, representation). My only concern is that self-consistency itself has a lot of limitations (e.g., requiring the answers to be short to be aggregated) and only has limited application scope, and so is the proposed approach.

The authors' rebuttal did a good job to address most of the concerns, and given the reviewers' unanimous positive scores, I lean acceptance of the paper.

**Reviewer Concerns:**

Concerns Addressed in rebuttal:

- Added empirical support tying proposed indicators (e.g., skew / cluster agreement measures) to observed gains
- Added analysis showing dominant clusters tend to be more accurate on average
- Added robustness experiments comparing multiple clustering algorithms and discussing hyperparameter choices.
- Expanded comparisons of TF-IDF vs. semantic embeddings, arguing semantic embeddings add overhead with limited benefit in this setting.
- Added one non-STEM benchmark

Concerns still outstanding:
- Generality remains limited: evidence beyond structured reasoning is still thin (one added non-STEM dataset helps a bit).

**Reviewer Scores:**

No reviewers explicitly mentions the score change, from my perspective:

- Reviewer w1dY (6 → 6)
- Reviewer yLoa (6 → 6)
- Reviewer jah9 (6 → 6 or 8): Could move slightly positive given added analyses (representation choice, validation of proposed indicators, failure investigation, and overhead discussion). If they still view generality as the main gap, they may remain at 6 but with higher confidence.
- Reviewer jggQ (6 → 6): Likely unchanged; they already found the approach promising but flagged applicability limits and sensitivity.

---

### Decision · Program_Chairs · 2026-01-26

Accept (Poster)